# Continual Learning with Bayesian Neural Networks for Non-Stationary Data

**Richard Kurle**[*1 2]      **Botond Cseke**[1]      **Alexej Klushyn**[1 2]

**Patrick van der Smagt**[1]      **Stephan Günnemann**[2]

[1]Volkswagen Group      [2]Technical University of Munich

## Abstract

This work addresses continual learning for non-stationary data, using Bayesian neural networks and memory-based online variational Bayes. We represent the posterior approximation of the network weights by a diagonal Gaussian distribution and a complementary memory of raw data. This raw data corresponds to likelihood terms that cannot be well approximated by the Gaussian. We introduce a novel method for sequentially updating both components of the posterior approximation. Furthermore, we propose Bayesian forgetting and a Gaussian diffusion process for adapting to non-stationary data. The experimental results show that our update method improves on existing approaches for streaming data. Additionally, the adaptation methods lead to better predictive performance for non-stationary data.

## 1 Introduction

Continual learning (CL), also referred to as lifelong learning, is typically described informally by the following set of desiderata for computational systems: the system should (i) learn *incrementally* from a data stream, (ii) exhibit *information transfer* forward and backward in time, (iii) avoid *catastrophic forgetting* of previous data, and (iv) *adapt* to changes in the data distribution (Ring, 1997; Silver et al., 2013; Chen & Liu, 2016; Ruvolo & Eaton, 2013; Parisi et al., 2018). The necessity to adapt to non-stationary data is often not reconcilable with the goal of preventing forgetting. This problem is also known as the stability-plasticity dilemma (Grossberg, 1987).

The majority of current CL research is conducted in the context of online multi-task learning (Nguyen et al., 2018; Kirkpatrick et al., 2017; Schwarz et al., 2018; Rusu et al., 2016; Fernando et al., 2017), where the main objective is to prevent catastrophic forgetting of previously learned tasks. This focus is reasonable since changes in the statistics of the data distribution are usually an artefact of learning different tasks sequentially. However, changes in the statistics of the data can also be real properties of the data-generating process. Examples include models of energy demand, climate analysis, financial market, or user-behavior analytics (Ditzler et al., 2015). In such applications, the statistics of the current data distribution are of particular interest. Old data may be outdated and can even deteriorate learning if the *drift* in the data distribution is neglected. Consequently, CL systems for non-stationary data require adaptation methods, which deliberately *forget* outdated information.

In this work, we develop an approximate Bayesian approach for training Bayesian neural networks (BNN) (Hinton & van Camp, 1993; Graves, 2011; Blundell et al., 2015) *incrementally* with non-stationary streaming data. Similar to variational continual learning (VCL) (Nguyen et al., 2018) and the Virtual Vector Machine (VVM) (Minka et al., 2009), we approximate the posterior using a Gaussian distribution and a complementary memory of previous data. Both components are *updated* sequentially, while *adapting* to changes in the data distribution. Our main contributions are as follows:

- We propose an online approximation consisting of a diagonal Gaussian distribution and a running memory, and we provide a novel sequential update method for both components.
- We extend the online approximation by two alternative adaptation methods, thereby generalising online variational Bayes with Bayesian neural networks to non-stationary data.

We compare our sequential update method to VCL in the online-inference setting on several popular datasets, demonstrating that our approach is favorable. Furthermore, we validate our adaptation methods on several datasets with *concept drift* (Widmer & Kubat, 1996), showing performance improvements compared to online variational Bayes without adaptation.

---

[*]Correspondence to `richard.kurle@tum.de`

## 2 BACKGROUND: ONLINE INFERENCE

Consider a stream of datasets $\{\mathcal{D}_{t_k}\}_{k=1}^K$, where $t_k$ are the time points at which datasets $\mathcal{D}_{t_k}$ are observed. For the moment, we assume that these datasets and the samples within are generated independently and identically distributed (i.i.d.). Methods for non-i.i.d. data are considered in Sec. 4.

In the Bayesian approach to online learning, we want to infer the posterior distribution $p(\mathbf{w}|\mathcal{D}_{t_1:t_k})$ of our model parameters, with the restriction that the data is processed sequentially.[1] Using Bayes rule, a recursive posterior inference equation emerges naturally:

$$p(\mathbf{w}|\mathcal{D}_{t_1:t_k}) \propto p(\mathbf{w}|\mathcal{D}_{t_1:t_{k-1}})\, p(\mathcal{D}_{t_k}|\mathbf{w}, \mathcal{D}_{t_1:t_{k-1}}) = p(\mathbf{w}|\mathcal{D}_{t_1:t_{k-1}})\, p(\mathcal{D}_{t_k}|\mathbf{w}), \qquad (1)$$

where the last step follows from the i.i.d. assumption of the data.

In this paper, we consider Gaussian and multinomial likelihoods, parametrised by a neural network with weights $\mathbf{w}$ and prior $p(\mathbf{w}|\emptyset) = p_0(\mathbf{w}) = \mathcal{N}(\mathbf{w}; \mu_0, \sigma_0)$. Furthermore, we consider supervised learning, where $\mathcal{D}_{t_k} = \{\mathbf{d}_{t_k}^{(n)}\}_n = \{(\mathbf{x}_{t_k}^{(n)}, \mathbf{y}_{t_k}^{(n)})\}_n$ and $p(\mathbf{d}_{t_k}^{(n)}|\mathbf{w}) = p(\mathbf{y}_{t_k}^{(n)} \,|\, \mathrm{NN}(\mathbf{x}_{t_k}^{(n)}; \mathbf{w}))$.

### 2.1 ONLINE VARIATIONAL BAYES

Since exact Bayesian inference is intractable for non-trivial models, various approximations have been developed. Prominent examples include sequential Monte Carlo (Liu & Chen, 1998), assumed density filtering (Maybeck, 1982), and online variational Bayes (Opper, 1998; Ghahramani, 2000; Sato, 2001; Broderick et al., 2013). Online variational Bayes (VB) approximates the posterior of Eq. (1) by a parametrised distribution $q_{\theta_{t_k}}(\mathbf{w}) \approx p(\mathbf{w}|\mathcal{D}_{t_1:t_k})$ through a sequence of projections:

$$q_{\theta_{t_k}}(\mathbf{w}) = \underset{q_\theta}{\operatorname{argmin}}\, \mathrm{KL}\big[q_\theta(\mathbf{w}) \,\|\, Z_{t_k}^{-1}\, q_{\theta_{t_{k-1}}}(\mathbf{w})\, p(\mathcal{D}_{t_k}|\mathbf{w})\big], \qquad (2)$$

where $Z_{t_k}$ is the normalisation constant. The above minimisation is equivalent to maximising the evidence lower bound (ELBO) $\mathcal{L}_{t_k}(\theta; \mathcal{D}_{t_k}) = \mathbb{E}_{q_\theta(\mathbf{w})}\big[\log p(\mathcal{D}_{t_k}|\mathbf{w})\big] - \mathrm{KL}\big[q_\theta(\mathbf{w}) \,\|\, q_{\theta_{t_{k-1}}}(\mathbf{w})\big]$. In this work, we consider diagonal Gaussian posterior approximations $q_{\theta_{t_k}}(\mathbf{w})$ for the neural network weights, similar to Nguyen et al. (2018).

### 2.2 ONLINE VARIATIONAL BAYES WITH MEMORY

Online approximate Bayesian inference methods inevitably suffer from an information loss due to the posterior approximation at each time-step. An alternative approach to online learning is to store and update a representative dataset/generative model—and to use it as a memory—in order to improve inference (Robins, 1995; Lopez-Paz & Ranzato, 2017; Shin et al., 2017; Kamra et al., 2017). Memory-based online learning has also been combined with online Bayesian inference methods (Minka et al., 2009; Nguyen et al., 2018). A common property of these approaches is to represent the (current) posterior approximation by a product of two factors

$$p(\mathbf{w}|\mathcal{D}_{t_1:t_k}) \approx q_{\theta_{t_k}}(\mathbf{w})\, p(\mathcal{M}_{t_k}|\mathbf{w}) \qquad (3)$$

and update them sequentially as new data $\mathcal{D}_{t_k}$ is observed. The factor $p(\mathcal{M}_{t_k}|\mathbf{w}) = \prod_m^M p(\mathbf{m}_{t_k}^{(m)}|\mathbf{w})$ is the likelihood of a set of $M = |\mathcal{M}|$ data points, which we refer to as *running memory*; and $q_{\theta_{t_k}}(\mathbf{w})$ is a Gaussian distribution, which summarises the rest of the data $\bar{\mathcal{D}}_{1:t_k} = \mathcal{D}_{1:t_k} \setminus \mathcal{M}_{t_k}$.

In case of VCL, the factors in Eq (3) are updated in two steps, which we refer to as (i) *memory update* and (ii) *Gaussian update*: (i) a new memory $\mathcal{M}_{t_k} \subset \mathcal{D}_{t_k} \cup \mathcal{M}_{t_{k-1}}$ is selected using heuristics such as *random* selection or the *k-center* method (a greedy algorithm that selects $K$ data points based on geometric properties of $\mathcal{D}_{t_k} \cup \mathcal{M}_{t_{k-1}}$.); (ii) the Gaussian distribution is updated with the remaining data $\bar{\mathcal{D}}_{t_k} = \mathcal{D}_{t_k} \cup \mathcal{M}_{t_{k-1}} \setminus \mathcal{M}_{t_k}$ (using Eq. (2)) to obtain $q_{\theta_{t_k}}(\mathbf{w}) \approx q_{\theta_{t_{k-1}}}(\mathbf{w})\, p(\bar{\mathcal{D}}_{t_k}|\mathbf{w})$.

Note that we cannot sample directly from the posterior approximation in Eq. (3) and thus we cannot easily evaluate quantities such as the posterior predictive distribution. VCL therefore performs a second projection

$$\tilde{q}_{\theta_{t_k}}(\mathbf{w}) = \underset{q_\theta}{\operatorname{argmin}}\, \mathrm{KL}\Big[q_\theta(\mathbf{w}) \,\|\, \tilde{Z}_{t_k}^{-1}\, q_{\theta_{t_k}}(\mathbf{w})\, p(\mathcal{M}_{t_k}|\mathbf{w})\Big]. \qquad (4)$$

This distribution should not be confused with the recursively updated variational distribution (Eq. (2)).

---

[1] A strict definition of online learning requires single data samples at each time step instead of batches $\mathcal{D}_{t_k}$.

## 3 IMPROVING MEMORY-BASED ONLINE VARIATIONAL BAYES

In this section, we focus on two problems of existing approaches using online VB with a running memory: (i) the *memory update* does not take into account the approximation error or approximation capabilities of the variational distribution; (ii) the *Gaussian update*—performed by optimising the ELBO (Eq. (2)) only with data $\bar{\mathcal{D}}_{t_k}$—can fail for streaming data. This is because VB yields poor posterior approximations if the dataset is too small or the neural network architecture has too much capacity (cf. Ghosh et al. (2018), Fig. 1). In Secs. 3.2 and 3.3, we propose improvements to these two update methods. The mathematical background for our approach is provided in Sec. 3.1.

### 3.1 PROPERTIES OF THE GAUSSIAN VARIATIONAL APPROXIMATION

There are two important properties of the Gaussian variational approximation that we will exploit later: (i) Gaussian approximate posterior distributions factorise into a product of Gaussian terms corresponding to the prior and each likelihood term; (ii) the ELBO can be written as the sum of the approximation's normalisation constant and a sum of residuals corresponding to these factors.

Let $p_0(\mathbf{w}) = \mathcal{N}(\mathbf{w}; \mu_0, \Sigma_0)$ be a Gaussian prior and $p(\mathcal{D}|\mathbf{w}) = \prod_n p(\mathbf{d}^{(n)}|\mathbf{w})$ be the likelihood of the observed data $\mathcal{D}$. Furthermore, let $q_\theta(\mathbf{w}) = \mathcal{N}(\mathbf{w}; \mu, \Sigma)$ denote the corresponding Gaussian variational approximation with $\theta = \{\mu, \Sigma\}$. Assume that $\mu$ and $\Sigma$ are the optimal parameters corresponding to a (local) maximum of the ELBO $\mathcal{L}(\mu, \Sigma; \mathcal{D})$. The optimality conditions $\partial_\mu \mathcal{L}(\mu, \Sigma; \mathcal{D}) = 0$ and $\partial_\Sigma \mathcal{L}(\mu, \Sigma; \mathcal{D}) = 0$ can be rewritten as follows (Knowles & Minka, 2011; Opper & Archambeau, 2008; Cseke et al., 2013) (cf. App. C):

$$\Sigma^{-1}\mu = \Sigma_0^{-1}\mu_0 + \sum_n \left( \partial_\mu \mathbb{E}_{q_\theta(\mathbf{w})}\big[ \log p(\mathbf{d}^{(n)}|\mathbf{w}) \big] - 2\partial_\Sigma \mathbb{E}_{q_\theta(\mathbf{w})}\big[ \log p(\mathbf{d}^{(n)}|\mathbf{w}) \big]\mu \right), \quad (5a)$$

$$\Sigma^{-1} = \Sigma_0^{-1} - 2\sum_n \partial_\Sigma \mathbb{E}_{q_\theta(\mathbf{w})}\big[ \log p(\mathbf{d}^{(n)}|\mathbf{w}) \big]. \quad (5b)$$

Since the sum of natural parameters corresponds to a product in distribution space, the above equations show that—at a local optimum—the approximation $q_\theta(\mathbf{w})$ factorises in the same way as the posterior $p(\mathbf{w}|\mathcal{D})$. It can be written in the form $q_\theta(\mathbf{w}) = Z_q^{-1} p_0(\mathbf{w}) \prod_n \mathbf{r}^{(n)}(\mathbf{w})$, where the factors $\mathbf{r}^{(n)}(\mathbf{w})$ are Gaussian functions with natural parameters given by Eqs. (5a) and (5b), and where $Z_q = \int p_0(\mathbf{w}) \prod_n \mathbf{r}^{(n)}(\mathbf{w}) \, d\mathbf{w}$ is the normalisation constant. These Gaussian functions $\mathbf{r}^{(n)}(\mathbf{w})$ each correspond to the contribution of the likelihood $p(\mathbf{d}^{(n)}|\mathbf{w})$ to the posterior approximation $q_\theta(\mathbf{w})$.

The resulting factorisation implies that the ELBO $\mathcal{L}(\mu, \Sigma; \mathcal{D})$ can be written in the form (Opper & Winther, 2005) (c.f. App. D)

$$\mathcal{L}(\mu, \Sigma; \mathcal{D}) = \log Z_q + \sum_n \mathbb{E}_{q_\theta(\mathbf{w})}\big[ \log p(\mathbf{d}^{(n)}|\mathbf{w}) - \log \mathbf{r}^{(n)}(\mathbf{w}) \big]. \quad (6)$$

If the terms $p(\mathbf{d}^{(n)}|\mathbf{w})$ were (diagonal) Gaussian in $\mathbf{w}$, they would each cancel with the corresponding (diagonal) Gaussian term, leaving only $\log Z_q$. Intuitively, the residual terms in Eq. (6) can be used to quantify the quality of the Gaussian approximation.

### 3.2 MEMORY UPDATE

The authors of VCL propose to use a memory to compensate the information loss resulting from the Gaussian approximation of the posterior distribution. However, their *memory update* is independent of the approximation error that is due to the chosen distributional family (diagonal Gaussian). An alternative *memory update*, which specifically targets the above mentioned information loss, has been introduced previously for VVM. Although the latter method was developed for expectation propagation in a (linear) logistic regression model—and is thus not directly applicable to online VB—we show that some of its properties can be transferred to the variational inference setting. The central idea is to replace the likelihood terms that can be well approximated by a Gaussian distribution by their Gaussian proxies $p(\mathbf{d}_{t_k}|\mathbf{w}) \approx \mathbf{r}_{t_k}(\mathbf{w}; \mathbf{d}_{t_k})$ resulting in $q_{\theta_{t_k}}(\mathbf{w})$; and retain the data corresponding to the rest of the likelihood terms in the memory. To score a *candidate* memory, Minka et al. (2009) proposed to maximise the KL divergence between the model given in the form of Eq. (3)

and a Gaussian posterior approximation, that is, maximise $\mathrm{KL}\big[\tilde{Z}_{t_k}^{-1} q_{\theta_{t_k}}(\mathbf{w})\, p(\mathcal{M}|\mathbf{w}) \,\|\, \tilde{q}_{\theta_{t_k}}(\mathbf{w})\big]$. However, this score function is intractable, because the expectation in the KL includes the likelihood $p(\mathcal{M}|\mathbf{w})$. In the following, we develop a tractable score function applicable to VB. Intuitively, we can use Eq. (6) to test how much $\mathcal{L}(\mu, \Sigma; \mathcal{D})$ changes if we replace the exact likelihood terms (of all data which is not contained in the *candidate* memory) by their Gaussian approximations.

To achieve this, we need to find Gaussian approximations for every data point in the *candidate* memory. We first approximate the posterior distribution using both $\mathcal{D}_{t_k}$ and $\mathcal{M}_{t_{k-1}}$:

$$\tilde{q}_{\theta_{t_k}}(\mathbf{w}) = \operatorname*{argmin}_{q_\theta} \mathrm{KL}\Big[ q_\theta(\mathbf{w}) \,\|\, \tilde{Z}_{t_k}^{-1} q_{\theta_{t_{k-1}}}(\mathbf{w})\, p(\mathcal{D}_{t_k}|\mathbf{w})\, p(\mathcal{M}_{t_{k-1}}|\mathbf{w}) \Big]. \tag{7}$$

Next, we use Eqs. (5a) and (5b) to calculate the natural parameters of all Gaussian terms. In practice, we estimate the natural parameters using (unbiased) Monte-Carlo estimators for the expectations. We have now available the likelihood terms and their Gaussian approximations. This allows us to write $\mathcal{L}(\theta_{t_k}; \mathcal{D}_{t_k} \cup \mathcal{M}_{t_{k-1}})$ in the form of Eq. (6):

$$\mathcal{L}(\theta_{t_k}; \mathcal{D}_{t_k} \cup \mathcal{M}_{t_{k-1}}) = \log Z_{q_{t_k}} + \sum_{\mathbf{d}_{t_k} \in \mathcal{D}_{t_k} \cup \mathcal{M}_{t_{k-1}}} \mathbb{E}_{\tilde{q}_{\theta_{t_k}}(\mathbf{w})}\big[ \log p(\mathbf{d}_{t_k}|\mathbf{w}) - \log \mathbf{r}_{t_k}(\mathbf{w}; \mathbf{d}_{t_k}) \big],$$

where $\mathbf{d}_{t_k}$ are the samples in $\mathcal{D}_{t_k} \cup \mathcal{M}_{t_{k-1}}$ and where $\mathbf{r}_{t_k}(\mathbf{w}; \mathbf{d}_{t_k})$ are the Gaussian approximation of the corresponding likelihood terms. Note that $\mathbf{r}_{t_k}$ does not only depend on $\mathbf{d}_{t_k}$, however, we omit the dependence on the remaining data for notational convenience.

If the likelihood $p(\mathbf{d}_{t_k}|\mathbf{w})$ is close to the Gaussian $\mathbf{r}_{t_k}(\mathbf{w}; \mathbf{d}_{t_k})$ in expectation w.r.t. the approximate posterior $q_{\theta_{t_k}}(\mathbf{w})$, then its contribution to $\mathcal{L}(\theta_{t_k}; \mathcal{D}_{t_k} \cup \mathcal{M}_{t_{k-1}})$ is small. Similarly, likelihood terms that cannot be well approximated by the respective Gaussian have a large contribution, and, hence, the corresponding data should be kept in the memory. For this reason, we propose the score function

$$S_{t_k}(\mathcal{M}; \mathcal{D}_{t_k} \cup \mathcal{M}_{t_{k-1}}) = \sum_{\mathbf{d}_{t_k} \in \mathcal{M}} \mathbb{E}_{\tilde{q}_{\theta_{t_k}}(\mathbf{w})}\big[ \log p(\mathbf{d}_{t_k}|\mathbf{w}) - \log \mathbf{r}_{t_k}(\mathbf{w}; \mathbf{d}_{t_k}) \big], \tag{8}$$

and the corresponding *memory update* $\mathcal{M}_{t_k} = \operatorname{argmax}_{\mathcal{M}} S_{t_k}(\mathcal{M}; \mathcal{D}_{t_k} \cup \mathcal{M}_{t_{k-1}})$. Note that since all residual terms are computed independently, the update results in selecting the top $M$ terms.

### 3.3 Gaussian Update

The *Gaussian update* follows from the *memory update* presented in the previous section: once the memory $\mathcal{M}_{t_k}$ has been selected, we update the Gaussian distribution with the approximations corresponding to the rest of the data $\mathcal{D}_{t_k} \cup \mathcal{M}_{t_{k-1}} \setminus \mathcal{M}_{t_k}$. We can update $q_{\theta_{t_k}}(\mathbf{w})$ in two equivalent ways:

$$q_{\theta_{t_k}}(\mathbf{w}) = q_{\theta_{t_{k-1}}}(\mathbf{w}) \prod_{\mathbf{d}_{t_k} \notin \mathcal{M}_{t_k}} \mathbf{r}_{t_k}(\mathbf{w}; \mathbf{d}_{t_k}), \quad \text{(9a)} \qquad q_{\theta_{t_k}}(\mathbf{w}) = \tilde{q}_{\theta_{t_k}}(\mathbf{w}) \,/ \prod_{\mathbf{d}_{t_k} \in \mathcal{M}_{t_k}} \mathbf{r}_{t_k}(\mathbf{w}; \mathbf{d}_{t_k}). \quad \text{(9b)}$$

Note again that the natural parameters of $\mathbf{r}_{t_k}(\mathbf{w}; \mathbf{d}_{t_k})$ are estimated using Monte Carlo and the products in the above equations imply a summation of the natural parameters. In order to reduce the variance of this sum of estimators, we use Eq. (9a) if $|\mathcal{D}_{t_k}| \leq |\mathcal{M}_{t_k}|$, and Eq. (9b) if $|\mathcal{D}_{t_k}| > |\mathcal{M}_{t_k}|$. Furthermore, we can compute the average bias from all natural parameter estimates (see App. C). We reduce the bias of our estimates by subtracting the average bias from all estimates. Note that a further option to update $q_{t_k}(\mathbf{w})$ would be to use VB on the data $\mathcal{D}_{t_k} \cup \mathcal{M}_{t_{k-1}} \setminus \mathcal{M}_{t_k}$ to compute the update $q_{\theta_{t_k}}(\mathbf{w}) \approx q_{\theta_{t_{k-1}}}(\mathbf{w})\, p(\mathcal{D}_{t_k} \cup \mathcal{M}_{t_{k-1}} \setminus \mathcal{M}_{t_k}|\mathbf{w})$. The latter approach is numerically more stable but computationally more expensive. It also turned out that it is less favorable to the update using Eq. (9a) or Eq. (9b) in case of small datasets $\mathcal{D}_{t_k}$, because VB applied to BNNs with small datasets often leads to a poor fit.

Previous work hypothesised that this problem is an artifact of the ELBO and not an optimisation problem (Trippe & Turner, 2018; Turner & M. Sahani, 2011). We provide further evidence in Fig. 1, where we infer the posterior of a Bayesian neural network with VB, using 70 and 100 data samples respectively and compare it to posterior inference with MCMC. In case of VB with 70 samples, the posterior approximation yields a model that is almost linear. These difficulties of posterior inference with variational Bayes are especially problematic in case of the streaming data setting, where the number of observations at each time-step is typically very small. The *Gaussian update* proposed above can alleaviate the problem of having to train BNNs with small datasets. Specifically, we have $N_{t_k} + M$ instead of $N_{t_k}$ data points to find a better optimum of the ELBO.

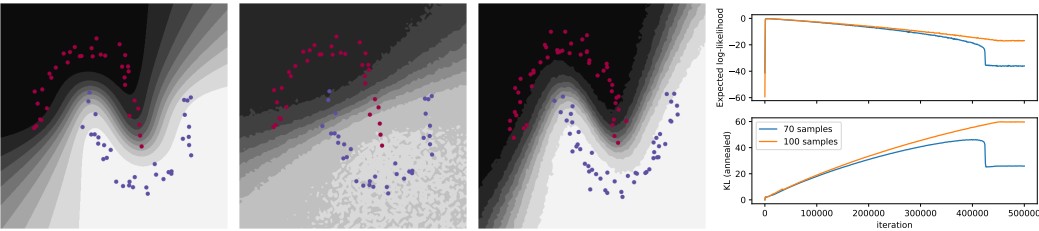

(a) MCMC, 70 samples    (b) VB, 70 samples    (c) VB, 100 samples    (d) ELBO terms during training

Figure 1: Posterior predictive distribution in the xy-plane (grey) of a Bayesian neural network with 2 layers of 16 units, tanh activations, prior $p_0(\mathbf{w}) = \mathcal{N}(\mathbf{w}; 0, 1)$, and Bernoulli likelihood. In case of variational Bayes (Figs. 1b, 1c), the KL divergence of the ELBO is annealed from $\beta = 0$ to $\beta = 1$ over many iterations (450k annealing, 50k ELBO). Fig. 1d shows that the approximation trades off the expected log-likelihood for a better KL divergence as $\beta$ is increased. With 70 data points, the annealed KL jumps to a significantly lower value, resulting in an almost linear decision boundary. By contrast, MCMC yields a much better predictive distribution for the same number of samples. Data is visualised in red and blue.

## 4   VARIATIONAL BAYES WITH MODEL ADAPTATION

The incremental learning methods discussed so far assume i.i.d. data (cf. Sec. 2, Eq. (1)). This assumption can be reasonable even in scenarios with changing data distributions, e.g. when the data drift is an algorithmic artifact rather than a real phenomenon. For example, in online multi-task or curriculum learning we want to learn a model of all tasks, but we may choose to learn the tasks incrementally for various reasons (e.g. Nguyen et al., 2018; Kirkpatrick et al., 2017; Schwarz et al., 2018; Rusu et al., 2016; Fernando et al., 2017). However, such approaches are not applicable for modeling non-stationary data: one of the properties of online VB is that the variance of the Gaussian posterior approximation shrinks at a rate of $O(N)$, where $N$ is the total amount of data (e.g. Opper, 1998). Consequently, learning comes to a halt as $t \to \infty$. To overcome this issue, the model needs to be extended by a method that enables it to adapt to changes in the data distribution, e.g., by deliberately forgetting the belief inferred from previous data.

In the following, we describe two alternative methods for adapting to changing data. In Sec. 4.1, we impose Bayesian exponential forgetting, which forgets previous data exponentially by weighting the likelihood terms (or their approximations). In Sec. 4.2, we implement the adaptation through a diffusion process applied to the neural network parameters. Compared to the online learning scenario, we make the following assumptions: (i) we observe datasets $\mathcal{D}_{t_k}$ at potentially non-equidistant time steps $t_k$; (ii) data within $\mathcal{D}_{t_k}$ is assumed i.i.d., however, not between different datasets $\mathcal{D}_{t_k}$ and $\mathcal{D}_{t_{k+1}}$.

In both approaches, we realise adaptation by an additional *forgetting step* before observing the new data $\mathcal{D}_{t_{k+1}}$. We denote the distribution, which results from applying the *forgetting step* to the posterior approximation $q_{\theta_{t_k}}(\mathbf{w}) \, p(\mathcal{M}_{t_k}|\mathbf{w})$ by $p_{t_{k+1}}(\mathbf{w})$.

### 4.1   ADAPTATION WITH BAYESIAN FORGETTING

Model adaptation through forgetting can be achieved by decaying the likelihood based on the temporal recency of the data (Graepel et al., 2010; Honkela & Valpola, 2003). It has been explored previously as an alternative to filtering and is referred to as Bayesian exponential forgetting (Kulhavý & Zarrop, 1993). This approach defines a forgetting operator that yields $p(\mathbf{w}_{t_{k+1}}|\mathcal{D}_{t_1:t_k})$ directly. Here, we use a continuous-time version of this forgetting operation that can be formulated as

$$p(\mathbf{w}|\mathcal{D}_{t_1:t_K}) \propto p_0(\mathbf{w}) \prod_{k=1}^{K} p(\mathcal{D}_{t_k}|\mathbf{w})^{(1-\epsilon)^{\frac{t_K - t_k}{\tau}}}, \qquad (10)$$

where $\tau$ is a time-constant corresponding to the average of the time-lags $\Delta t_{k+1} = t_{k+1} - t_k$. The distribution defined in Eq. (10) can be formulated recursively (cf. App. F) as

$$p(\mathbf{w}|\mathcal{D}_{t_1:t_{k+1}}) \propto p_0(\mathbf{w})^{1-(1-\epsilon)^{\Delta t_{k+1}/\tau}} p(\mathbf{w}|\mathcal{D}_{t_1:t_k})^{(1-\epsilon)^{\Delta t_{k+1}/\tau}} p(\mathcal{D}_{t_{k+1}}|\mathbf{w}). \qquad (11)$$

This equation can be viewed as Bayes rule (Eq.(1)) applied after the *forgetting step*. The first two terms of Eq. (11) can be identified as the forgetting operation, applied to the current posterior. In

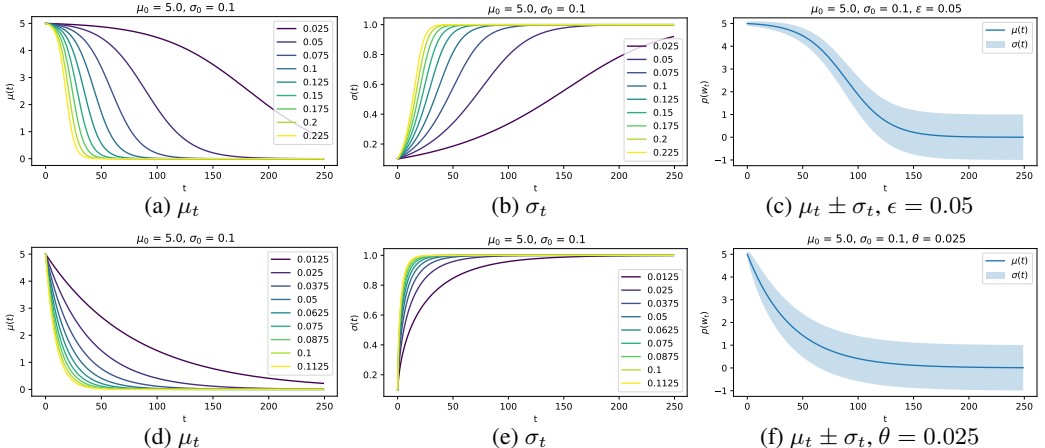

Figure 2: Time-evolution of distribution parameters of Bayesian Forgetting (top) and the Ornstein-Uhlenbeck process (bottom) for different adaptation parameter values. The initial distribution (at $t = 0$) can be seen as the approximate posterior at some time-step $t_k$.

order to apply this operation to our posterior approximation $q_{\theta_{t_k}}(\mathbf{w})\, p(\mathcal{M}_{t_k}|\mathbf{w})$, we modify it by an additional weighting factor for each likelihood term in the memory. Denoting the age of a memory item $\mathbf{m}$ by $\Delta t_k(\mathbf{m})$, the forgetting operation for this new posterior approximation then results in

$$p_{t_{k+1}}(\mathbf{w}) \propto p_0(\mathbf{w})^{1-(1-\epsilon)^{\Delta t_{k+1}/\tau}} \times \left[ q_{\theta_{t_k}}(\mathbf{w}) \prod_{\mathbf{m}\in\mathcal{M}_{t_k}} p(\mathbf{m}|\mathbf{w})^{(1-\epsilon)^{\Delta t_k(\mathbf{m})/\tau}} \right]^{(1-\epsilon)^{\Delta t_{k+1}/\tau}}$$

$$= \left[ p_0(\mathbf{w})^{1-(1-\epsilon)^{\Delta t_{k+1}/\tau}} q_{\theta_{t_k}}(\mathbf{w})^{(1-\epsilon)^{\Delta t_{k+1}/\tau}} \right] \times \prod_{\mathbf{m}\in\mathcal{M}_{t_k}} p(\mathbf{m}|\mathbf{w})^{(1-\epsilon)^{\Delta t_{k+1}(\mathbf{m})/\tau}}, \quad (12)$$

where $\Delta t_{k+1}(\mathbf{m}) = \Delta t_k(\mathbf{m}) + \Delta t_{k+1}$. As can be seen from Eq. (12), BF acts on both factors of the posterior approximation independently: in case of the memory, it re-weights the respective likelihood terms by updating $\Delta t_{k+1}(\mathbf{m})$. For the Gaussian term $q_{\theta_{t_k}}(\mathbf{w})$, BF leads to a weighted product with the prior distribution (i.e. the first two terms of Eq. (12)), resulting in a Gaussian with parameters

$$\sigma_{t_{k+1}}^{-2} = \left(1 - (1-\epsilon)^{\Delta t_{k+1}/\tau}\right)\sigma_0^{-2} + (1-\epsilon)^{\Delta t_{k+1}/\tau}\sigma_{t_k}^{-2},$$

$$\sigma_{t_{k+1}}^{-2}\mu_{t_{k+1}} = \left(1 - (1-\epsilon)^{\Delta t_{k+1}/\tau}\right)\sigma_0^{-2}\mu_0 + (1-\epsilon)^{\Delta t_{k+1}/\tau}\sigma_{t_k}^{-2}\mu_{t_k}.$$

For $\Delta t_{k+1} \to \infty$, the likelihood term in Eq. (12) converges to the uniform distribution and the Gaussian term reverts to the prior. We note, however, that while Eq. (11) is an exact recursive form of Eq. (10), the online VB approximation of Eq. (11) is not generally identical to the (offline) VB approximation of Eq. (10) due to its successive approximations. For tuning the hyperparameter $\epsilon$, we note that the weighting of likelihood terms corresponds to an effective dataset size of $1/\epsilon \cdot N$ (if all datasets are of equal size $N$). In Fig. 2, we also visualise the forgetting operation applied to the Gaussian part of the posterior approximation for multiple values of $\epsilon$.

## 4.2 Adaptation with Diffusion Processes

Model adaptation can also be realised by using dynamic model parameters that evolve according to a stochastic process. In this case, adaptation is achieved by the stochastic transition $p_{t_{k+1},t_k}(\mathbf{w}'|\mathbf{w})$ resulting in a prediction distribution

$$p_{t_{k+1}}(\mathbf{w}') = \int p_{t_{k+1},t_k}(\mathbf{w}'|\mathbf{w})\, p(\mathbf{w}|\mathcal{D}_{t_1:t_k})\, d\mathbf{w}, \quad (13)$$

where we consider Gaussian transitions $p_{t_{k+1},t_k}(\mathbf{w}'|\mathbf{w})$. However, this operation is generally not tractable for our posterior approximation $q_{\theta_{t_k}}(\mathbf{w})\, p(\mathcal{M}_{t_k}|\mathbf{w})$. Moreover, the forgetting operation implied by the transition does not retain the product form as in the case of BF. For this reason, we consider only a Gaussian posterior approximation (without memory) for this approach, that is $p_{t_{k+1}}(\mathbf{w}') = \int p_{t_{k+1},t_k}(\mathbf{w}'|\mathbf{w})\, q_{\theta_{t_k}}(\mathbf{w})\, d\mathbf{w}$.

As mentioned in Sec. 4.1, BF yields the prior distribution for $\Delta t_{k+1} \to \infty$. This is a desirable property, since it corresponds to forgetting all information conveyed by the data. In case of a Gaussian prior, the only Gaussian process that fulfills this requirement is the *Ornstein-Uhlenbeck* (OU) process given by the stochastic differential equation $d\mathbf{w}_t = \theta \cdot (\mu_0 - \mathbf{w}_t)\,dt + \sigma_0\sqrt{2\theta}\,dW_t$, where $\theta$ is the stiffness parameter which controls the drift rate towards $\mu_0$. To decouple the adaptation parameter from the rate at which data is observed, we rescale the stiffness parameter as $\theta = {a}/{\tau}$. The resulting prediction distribution $p_{t_{k+1}}(\mathbf{w}) = \mathcal{N}\big(\mu_{t_{k+1}},\ \sigma_{t_{k+1}}^2\big)$ is defined by the parameters

$$\mu_{t_{k+1}} = \big(1 - e^{-a\frac{\Delta t_{k+1}}{\tau}}\big)\mu_0 + e^{-a\frac{\Delta t_{k+1}}{\tau}}\mu_{t_k},$$

$$\sigma_{t_{k+1}}^2 = \big(1 - e^{-2a\frac{\Delta t_{k+1}}{\tau}}\big)\sigma_0^2 + e^{-2a\frac{\Delta t_{k+1}}{\tau}}\ \sigma_{t_k}^2.$$

An interesting observation is that both parameters evolve independently of each other. In contrast to BF, the mean and variance—instead of the natural parameters—follow an exponential decay. The hyperparameter $a$ can be determined e.g. through the half-time of the exponential decay of the mean parameter, given as $\tau_{1/2} = {1}/{\theta}$. We visualise the time evolution of the above parameters in Fig. 2.

## 5 Related Work

There are many Bayesian approaches to online learning, which differ mostly in the approximation of the posterior distribution at each time-step. Sequential Monte Carlo (Liu & Chen, 1998) approximates the posterior by a set of particles. Assumed Density Filtering (ADF) (Maybeck, 1982) and Bayesian online learning (Opper, 1998) are deterministic posterior approximations based on moment matching. Other deterministic approaches are based on Laplace's approximation (MacKay, 1992): Kirkpatrick et al. (2017) use multiple diagonal Gaussian posterior approximations of previous time-steps to regularise future tasks; Ritter et al. (2018) use a single (block-diagonal) posterior approximation, summarising all previous time-steps. The latter method is closer to Bayesian online inference, as it is an approximation of Eq. (1). Our work is based on online VB (Opper, 1998; Ghahramani, 2000; Sato, 2001; Broderick et al., 2013), which approximates the posterior at every time-step by minimising the KL-divergence between a parametric (here Gaussian) and the true posterior distribution. In contrast to online VB, we approximate the posterior by a Gaussian distribution and a running memory.

Other approaches are based on various types of episodic memory, motivated by their empirical success in preventing catastrophic forgetting. The basic idea of rehearsal (Ratcliff, 1990) is to train on both the new data and a subset of previous data or pseudo samples (Robins, 1995; Shin et al., 2017; Kemker & Kanan, 2017) sampled from a generative model. The memory-based online inference methods most similar to our approach are VCL (Nguyen et al., 2018) and VVM (Minka et al., 2009). Both methods use a Gaussian distribution and a running memory to approximate the posterior. VCL uses heuristics such as *random selection* or the *k-center method* to update the memory. However, both heuristics select the memory independently of the Gaussian approximation. By contrast, VVM updates the memory with data that cannot be well approximated by the Gaussian distribution. VVM uses expectation propagation for the posterior approximation in a logistic regression model and, therefore, it is not directly applicable to our work. We transferred the main idea of VVM to online VB and developed the corresponding *memory update* method. In our case, the memory is updated with data for which the ELBO changes most if the corresponding likelihood functions are approximated by a Gaussian. In contrast to these two approaches, we extend our model by an *adaptation method* that allows to cope with non-stationary data.

Many adaptation methods were developed in the context of *concept drift*, however, few of these approaches operate in the Bayesian framework. For example McInerney et al. (2015) treat the learning dynamics of their model as a non-stationary process that allows for adaptation. In contrast, our approach uses an evolving prior and a well defined forgetting mechanism that gives a better control over the learning process. A more closely related approach uses the extended Kalman filter to estimate the optimal parameters of a logistic regression classifier Su et al. (2008). However, they consider a transition model which is equivalent to a Wiener process (in unit-time) and therefore does not revert to the prior. By contrast, our approach (Sec. 4.2) models the dynamics as a prior-reverting OU process. BF (Kulhavý & Zarrop, 1993) has been applied as an alternative to adaptation with an explicit transition model (e.g. Honkela & Valpola, 2003; Graepel et al., 2010). Compared to previous work, we used a continuous-time version of BF and extended it to our posterior appoximation consisting of a Gaussian and a running memory.

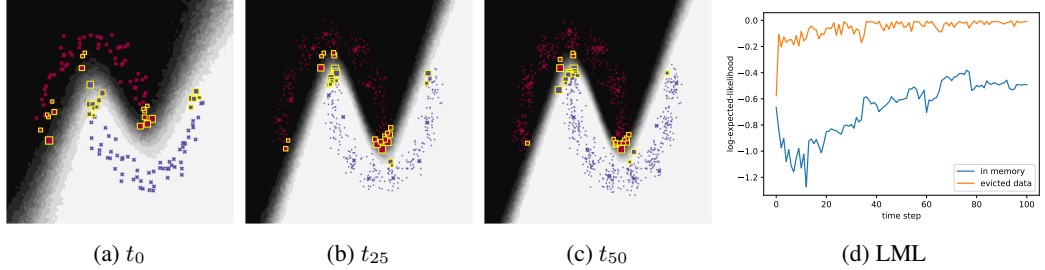

(a) $t_0$      (b) $t_{25}$      (c) $t_{50}$      (d) LML

Figure 3: Two-moons dataset. Predictive distribution (Figs. 3a – 3c) of a BNN (gray) and running memory (rectangular shapes, size is proportional to the score), chosen by the *memory update* proposed in Sec. 3.2. Data from $t_k$ and $t_{<k-1}$ is visualised as large circles and small dots, respectively. Fig. 3d shows the one-step-ahead (predictive) LML (divided by the number of samples) for data that will be selected for the memory and data that will be evicted. Data that will be selected in the memory tends to have a significantly lower predictive likelihood.

|         | GRS (ours)              | k-center           | random             |
|---------|-------------------------|--------------------|--------------------|
| Concrete| $\mathbf{-0.779 \pm 0.039}$ | $-0.798 \pm 0.039$ | $-0.800 \pm 0.039$ |
| Boston  | $\mathbf{-0.619 \pm 0.111}$ | $-0.638 \pm 0.093$ | $-0.664 \pm 0.156$ |
| Energy  | $\mathbf{0.365 \pm 0.440}$  | $-0.119 \pm 0.128$ | $-0.078 \pm 0.087$ |
| Yacht   | $\mathbf{1.925 \pm 0.229}$  | $1.658 \pm 0.291$  | $1.648 \pm 0.254$  |
| Spam    | $\mathbf{-0.216 \pm 0.016}$ | $-0.219 \pm 0.015$ | $-0.217 \pm 0.016$ |
| Wine    | $\mathbf{-1.165 \pm 0.056}$ | $-1.212 \pm 0.059$ | $-1.194 \pm 0.070$ |
| MNIST   | $\mathbf{-0.148 \pm 0.005}$ | $-0.158 \pm 0.005$ | $-0.153 \pm 0.005$ |

Table 1: average test LML, averaged over the last $10\%$ time-steps. Mean and std. deviations are computed over 16 independent runs. The memory size is 15 for Concrete, Boston, Energy and Yacht, 25 for Spam and Wine, and 150 for MNIST. Bold indicates best (average) results.

## 6 EXPERIMENTS

We validate our proposed inference methods in two stages. In Sec. 6.1, we compare our *memory update* and *Gaussian update* (Sec. 3) to existing memory-based online inference methods on several standard machine learning datasets. In Sec. 6.2, we evaluate our *adaptation methods* (Sec. 4) on commonly used datasets with *concept drift* (Widmer & Kubat, 1996), where the conditional distribution of labels given the features changes over time (i.e. non-stationary data in the context of predictive models).

We found that training (variational) Bayesian neural networks on streaming data is challenging, specifically, our approach requires model parameters very close to a local optimum since Eqs. (5a) and (5b) hold only at local extrema of the ELBO. To overcome these difficulties, we use several methods to reduce the variance of the gradient estimates for learning: (i) we apply the local reparametrisation trick (Kingma et al., 2015); (ii) we use the Adam optimiser (Kingma & Ba, 2014); and (iii) we use multiple Monte Carlo samples to estimate the gradients (cf. Tab. 2 for details). Furthermore, we developed methods for determining hyperparameters of the Gaussian prior and the initialisation distribution of Bayesian neural networks. The idea is similar to the initialisation method proposed by Glorot & Yoshua Bengio (2010) and He et al. (2015): we choose the prior and the posterior initialisation such that the mean and standard deviation of the activations in every layer are approximately zero and one, respectively. We refer to App. H and App. I for a derivation and further details.

We use the following metrics for evaluation: (i) the avg. test log-marginal likelihood (LML) $N_{\text{test}}^{-1} \sum_n \log \mathbb{E}_{\tilde{q}_{\theta_{t_k}}(\mathbf{w})} \left[ p(\mathbf{d}_{\text{test}}^{(n)} | \mathbf{w}) \right]$, where $\mathbf{d}_{\text{test}}^{(n)}$ is a sample from a heldout test dataset; (ii) the avg. one-step-ahead LML $N_{t_{k+1}}^{-1} \sum_n \log \mathbb{E}_{\tilde{q}_{\theta_{t_k}}(\mathbf{w})} \left[ p(\mathbf{d}_{t_{k+1}}^{(n)} | \mathbf{w}) \right]$, where $\mathbf{d}_{t_{k+1}}^{(n)}$ is data observed at time-step $t_{k+1}$. Both metrics measure the predictive performance, however (i) can be used in the online setting, where the data is i.i.d.; and (ii) is typically used to evaluate models with non-stationary streaming data.

### 6.1 ONLINE LEARNING

In this section, we evaluate our running memory (Sec. 3) in an online learning setting. To illustrate how our *memory update* works, we start our evaluation with a qualitative assessment: we train a model on 2-dimensional toy data (two-moons), where we can visualise the selected memory. The BNN has 2 layers with 16 units and $tanh$ activations, and has a prior $p_0(\mathbf{w}) = \mathcal{N}(\mathbf{w}; 0, 1)$ on all

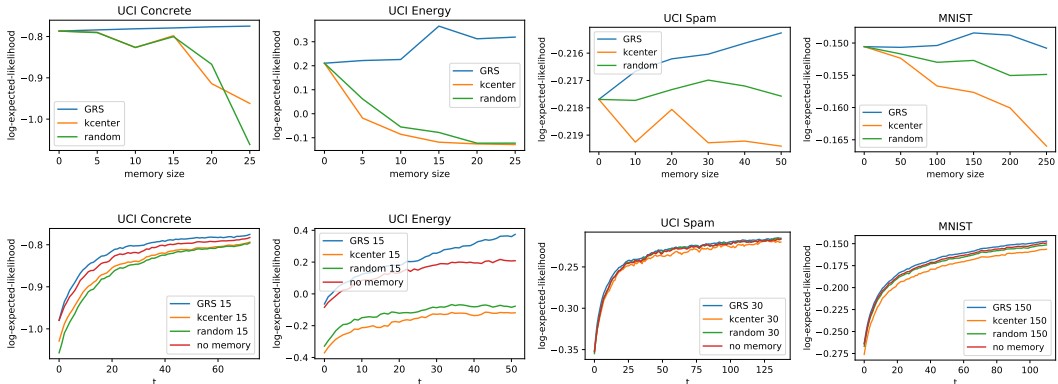

Figure 4: Average test LML, evaluated for several memory sizes (top), and evaluated over time (bottom) for a specific memory size (cf. corresponding legend). Cf. Sec. 6.1 for details and App. A for further results.

weights and biases. The memory-size is $M = 30$. The model observes 150 data samples at time-step $t_0$ and 15 samples at all consecutive time-steps. In Fig. 3, we visualise the selected memory and the corresponding scores for time-steps $t_0$, $t_{25}$, and $t_{50}$, respectively. We can make the empirical observation that our method favors data close to the decision boundary. Furthermore, in Fig. 3d, we visualise the one-step-ahead LML for data that will be selected and evicted (in the next time-step), respectively. This shows that our *memory update* tends to select data for which the model has a low predictive LML. These observations support our intuition that the memory is indeed complementary to the Gaussian approximation, selecting data for which the likelihood cannot be well approximated by a Gaussian function. In Fig. 9 of the supplementary material, we visualised the running memory for a model trained on MNIST, showing that the memory also accumulates diverse samples over time.

We evaluate our memory-based online inference method (Sec. 3) quantitatively on several standard machine learning datasets, including regression (UCI Boston, UCI Concrete, UCI Energy, UCI Yacht) and classification (MNIST, UCI Spam, UCI Wine) tasks. Here, we refer to our approach as Gaussian Residual Scoring (GRS). We compare GRS to the respective *memory update* and *Gaussian update* methods proposed in VCL (Nguyen et al., 2018) (cf. Sec. 2.2). Refer to App. B for an explanatory list of compared update methods. Online learning is performed by observing $N_{t_k}$ samples per time-step (cf. Tab. 2 for the experiment setup and hyperparameters.). For evaluation, we use a random held-out test dataset (20% of the data). We perform each experiment with 16 different random data splits and random seeds for the model parameter initialisation. In Fig. 4, we plot the test LML, averaged over the 16 runs, against the memory size, and the LML over all time-steps. In most cases, *random selection* and the *k-center* method start with a worse initial fit at $t_0$. This is because these methods perform the initial *Gaussian update* by optimising the ELBO with $N_{t_0} - M$ samples at $t_0$; by contrast, *GRS* uses a *Gaussian update* that first optimises the ELBO with $N_{t_0}$ samples and subsequently discounts the contribution of the memory. In Tab. 1, we report the mean and std. deviation of the LML, where the mean and std. deviation are taken over the 16 independent runs, each averaged over the last 10% time-steps. The results demonstrate the superior predictive performance of our update methods. We also note that the experiments on the smaller datasets (cf. Tab. 2 in App. B) result in a high variance among the random data splits and random seeds. This is the case for all compared methods and it could not be remedied e.g. by using annealing or a different prior.

## 6.2 ADAPTATION

In this section, we evaluate our adaptation methods (Sec. 4) in settings with *concept drift*. We begin with a simple logistic regression problem, where the data $\mathcal{D}_{t_k} = \{(\mathbf{x}_{t_k}, \mathbf{y}_{t_k})\}_n$, $\mathbf{x}_{t_k} \in \mathbb{R}^2$, $\mathbf{y}_{t_k} \in \mathbb{R}$ is sampled from $\mathbf{x}_{t_k} \sim \text{Uniform}(-3, 3)$, $\mathbf{y}_{t_k} \sim \text{Bernoulli}(\sigma(\mathbf{w}_{t_k} \mathbf{x}_{t_k}))$. The true model has two time-dependent parameters $\mathbf{w}_{t_k}^0 = 10 \sin(\alpha \cdot t_k)$, $\mathbf{w}_{t_k}^1 = 10 \cos(\alpha \cdot t_k)$, where $\alpha = 5 \deg / \sec$ and where we observe data at $t_k \in [0, 1, \ldots, 720]$. Fig. 5 shows the learned model parameters for standard online learning (without adaptation), OU process transitions, and Bayesian forgetting. If the time-dependence of the data is ignored (in case of online VB), the class labels are distributed with equal probability in the whole input space. Consequently, as $t \to \infty$, the weights of the model without adaptation shrink to 0. By contrast, the posterior means of BF and the OU process follow a sinusoidal curve as the parameters of the true model.

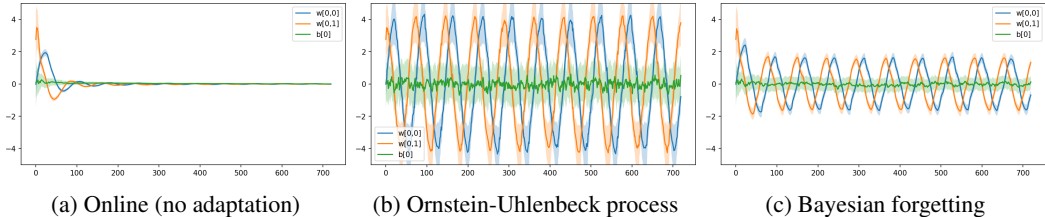

|     |     |     |
| :-: | :-: | :-: |
| (a) Online (no adaptation) | (b) Ornstein-Uhlenbeck process | (c) Bayesian forgetting |

Figure 5: Mean and std deviation of the approximate posterior distributions of a logistic regression model over 720 time-steps. The model is trained on a toy classification problem with rotating class boundaries (cf. Sec. 6.2). Online VB (left) quickly converges to zero mean, whereas Bayesian forgetting and OU-process transitions lead to a sinusoidal curve as in the true model.

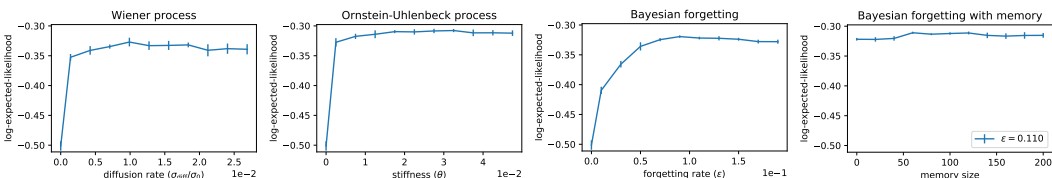

Figure 6: One-step ahead LML on Covertype dataset. Subplots show 3 different adaptation methods (3 left plots), evaluated for several values of the respective adaptation parameter, and Bayesian forgetting with $\epsilon = 0.11$, evaluated for multiple memory sizes (right).

We also evaluate our adaptation methods quantitatively on 3 datasets with *concept drift* (Weather, Gas Sensor Array Drift, Covertype). We compare online VB (without adaptation), the Wiener process (a special case of the OU process), the OU process, and Bayesian Forgetting (with and without memory). All compared variants use the same model architecture and hyperparameters (cf. Tab. 2 in the supplementary material). We report the one-step-ahead LML, where the expectation is approximated with 500 Monte Carlo samples. Results are averaged over the last 50% time-steps, because we are interested in the continual learning performance, and the first few time-steps will be similar for most methods. We report the mean and std. deviation over 8 independent runs with different random seeds. In Fig. 6 (and Fig. 10 in the appendix), we plot the LML against 10 adaptation parameter values (of the respective adaptation method), where the value zero corresponds to online VB. The LML for BF with different memory sizes and a fixed forgetting rate $\epsilon = 0.11$ is shown in Fig. 6. As can be seen from the results, all adaptation methods significantly improve the performance compared to online VB. Interestingly, the Ornstein-Uhlenbeck process performs better than Bayesian Forgetting, however, using a running memory with Bayesian Forgetting closes the gap.

## 7 CONCLUSION

In this work, we have addressed online inference for non-stationary streaming data using Bayesian neural networks. We have focused on posterior approximations consisting of a Gaussian distribution and a complementary running memory, and we have used variational Bayes to sequentially update the posteriors at each time-step. Existing methods update these two components without having an interaction between them, and they lack methods to adapt to non-stationary data. We have proposed a novel update method, which treats both components as complementary, and two novel adaptation methods (in the context of Bayesian neural networks with non-stationary data), which gradually revert to the prior distribution if no new data is observed.

Future research could extend our work by drift detection methods and use them to infer the adaptation parameters. This work could also be extended by developing adaptation methods for gradual, abrupt, or recurring changes in the data distribution. Finally, we observed that variational Bayesian neural networks with a uni-modal approximate posterior often find poor local minima if the dataset is small and models are complex. This is especially challenging in scenarios with streaming data. While our Gaussian update alleviates this problem to a certain degree, further research in extending the approximation family beyond Gaussians could be beneficial. Progress in this direction would improve our proposed methods and allow to scale them to more complex models.

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

## 8 APPENDIX

### A FURTHER EXPERIMENTAL RESULTS

#### A.1 MEMORY

Here we provide additional experimental results for the *memory update* and *Gaussian update* from Sec. 3. We conducted experiments on 3 additional datasets (UCI Boston, UCI Yacht, UCI Red Wine). The influence of the memory size and the performance over time (for a specific memory size) are shown in Fig. 7 (corresponding to Fig. 4 in the main text).

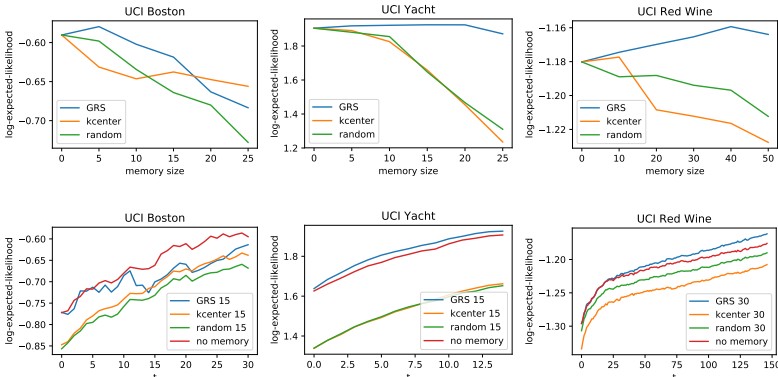

Figure 7: Average test LML on further datasets not included in the main text. Evaluated for several memory sizes (top), and evaluated over time (bottom) for a specific memory size. Cf. Sec. 6.1, App. A.1 for details.

Furthermore, we test the *memory update* and *Gaussian update* of GRS separately on UCI Energy and UCI Concrete. For this purpose, i) we combine the k-center method with our *Gaussian update* from Sec. 3.3; and ii) we use our *memory update* from Sec. 3.2 and update the Gaussian distribution by optimizing Eq. (2) with $\mathcal{D}_{t_k} \cup \mathcal{M}_{t_{k-1}} \backslash \mathcal{M}_{t_k}$ (re-fitting) . The results are shown in Fig. 8. As can be seen, GRS performs better than one of the components used in combination with a baseline method. *GRS with refit* performs especially bad, similar to the baselines *k-center* and *random*. This is because refitting requires optimising the ELBO with a small dataset. As mentioned in Sec. 3.3 (cf. Fig. 1), Bayesian neural networks with VB perform bad on small datasets due to over-regularisation. Consequently, in case of refitting, a good *memory update* can lead to a worse overall performance due to a much worse *Gaussian update*. While this general issue with Bayesian neural networks (learned with VB) is beyond the scope of this work, it is an important future research direction.

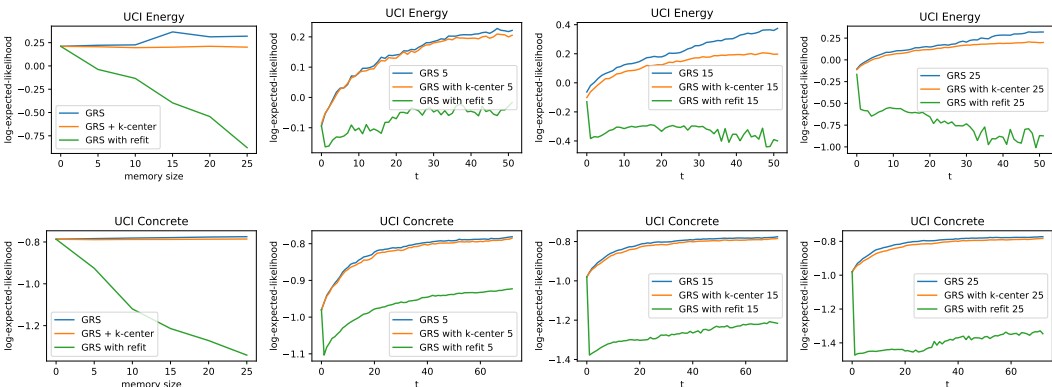

Figure 8: Average test LML on UCI Energy (top) and UCI Concrete (bottom). *GRS* denotes our approach (Sec. 3), *GRS with k-center* replaces our *memory update* by the k-center method; *GRS with refit* replaces our *Gaussian update* by the optimization of Eq. (2) with $\mathcal{D}_{t_k} \cup \mathcal{M}_{t_{k-1}} \backslash \mathcal{M}_{t_k}$. Evaluated for several memory sizes (left), and evaluated over time (right) for 3 different memory sizes. Hyperparameters are chosen as in Sec. 6.1.

To better understand our *memory update* using the score function from Eq. (8), we visualise the running memory for a model trained on MNIST in Fig. 9.

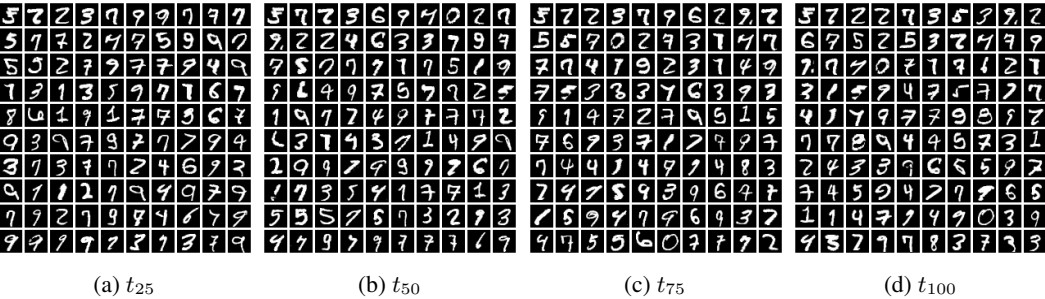

(a) $t_{25}$         (b) $t_{50}$         (c) $t_{75}$         (d) $t_{100}$

Figure 9: Running memory at different time-steps on MNIST (cf. Sec. 6.1), with a memory size $N = 100$. The *memory update* tends to select non-typical data while showing diversity.

## A.2    ADAPTATION

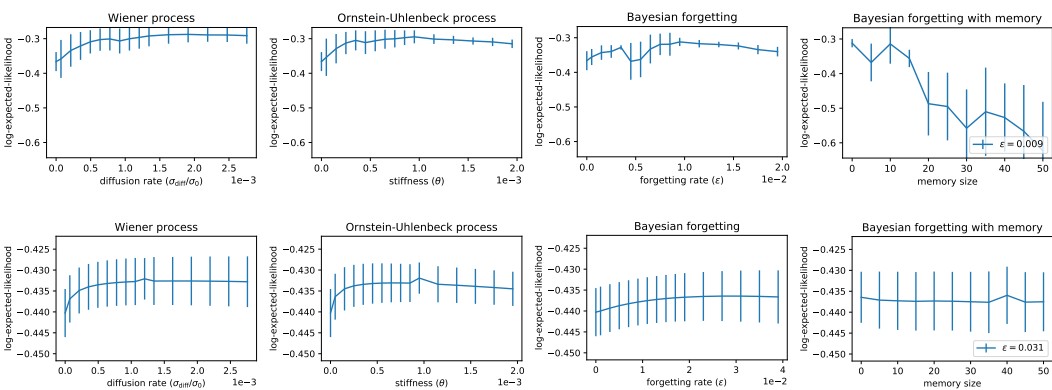

Figure 10: One-step ahead LML on Gas Sensor Array Drift dataset (top) and Weather dataset (bottom). Subplots show 3 different adaptation methods (left), evaluated for several values of the respective adaptation parameter, and Bayesian forgetting with $\epsilon = 0.0095$ (Gas Sensory Array Drift) and $\epsilon = 0.031$ (Weather), evaluated for multiple memory sizes (right).

We also evaluated our adaptation methods on 2 additional datasets (Gas Sensor Array Drift, Weather). In Fig. 10, we visualise the influence of the adaptation parameter for these datasets. Note that the range of the adaptation parameters is on a much smaller range compared to the experiments on Covertype (Sec. 6.2). For larger values, the performance starts to degrade. Surprisingly, the memory degrades the performance in case of the Gas Sensor Array Drift dataset.

## A.3    CATASTROPHIC FORGETTING WITH ONLINE VB AND BAYESIAN NEURAL NETWORKS

Here we provide further experimental results for the behavior of online VB (Secs. 2.1, 3) in case of non-stationary data. For this purpose, we train Bayesian neural networks with different architectures on the toy classification problem with a rotating decision boundary from Sec. 6.2, however, with 150 data samples per time-step. In Fig. 11, we visualise the training LML for different architectures, including a linear model. It can be seen that Bayesian neural networks with higher complexity (i.e. more layers or more units) drop slower in performance compared to the linear model. However, this is not a desired property for online VB, since exact online Bayesian inference would yield the same posterior distribution as offline Bayesian inference. In case of our toy classification data (where the time dependence is ignored), online inference should not be able to classify the data as $t \rightarrow \infty$. Instead, this learning behavior shows that online VB with Gaussian approximate posterior distributions is prone to catastrophic forgetting.

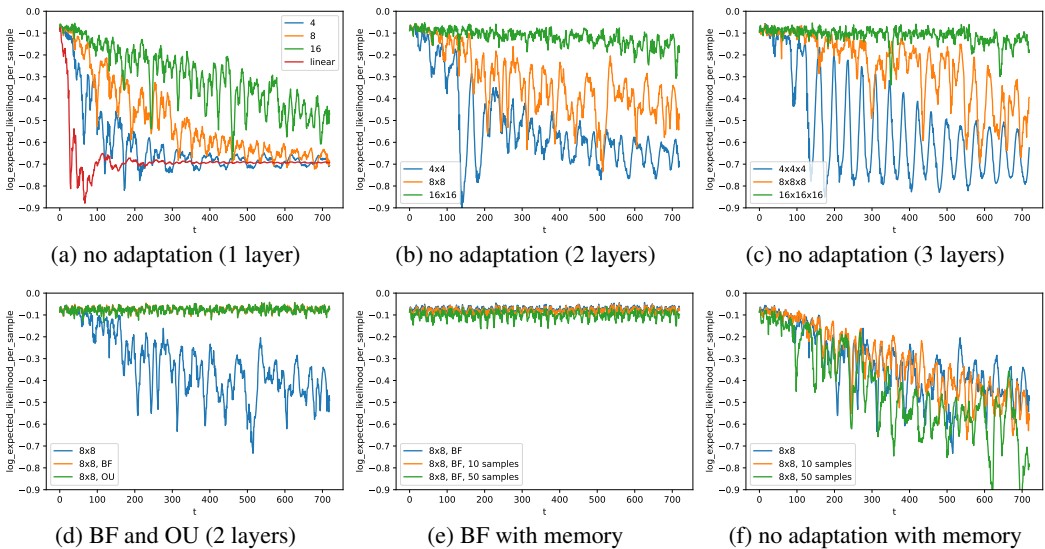

Figure 11: LML for toy classification problem with rotating class boundaries (cf.Sec. A)

## B EXPERIMENT SETUP

The following is an explanatory list of the update methods used in Sec. 6.1:

- k-center (VCL): Uses the k-center method (Sec.2.2) for the *memory update* and Eq. (2) with $(\mathcal{D}_{t_k} \cup \mathcal{M}_{t_{k-1}}) \setminus \mathcal{M}_{t_k}$ for the *Gaussian update*.

- random (VCL): Uses random selection (Sec.2.2) for the *memory update* and Eq. (2) with $(\mathcal{D}_{t_k} \cup \mathcal{M}_{t_{k-1}}) \setminus \mathcal{M}_{t_k}$ for the *Gaussian update*.

- GRS (Gaussian Residual Scoring, ours): Uses Eq. (8) for the *memory update* (Sec. 3.2) and performs the *Gaussian update* by first using Eqs. (2) with $(\mathcal{D}_{t_k} \cup \mathcal{M}_{t_{k-1}})$ and subsequently using Eqs. (5a), (5b) for removing the local contributions of $\mathcal{M}_{t_k}$ (cf. Sec. 3.3).

Similarly, the following list summarises the adaptation methods used in Sec. 6.2:

- Wiener process: Posterior approximation consists of $q_{\theta_{t_k}}(\mathbf{w})$ only. Transition $p(\mathbf{w}_{t_{k+1}}|\mathbf{w}_{t_k})$ is given by a *random walk*. We used a diffusion that is proportional to the prior standard deviation in every neural network layer (cf. Sec. 4.2). No memory used.

- Ornstein-Uhlenbeck process: Posterior approximation consists of $q_{\theta_{t_k}}(\mathbf{w})$ only. Transition $p(\mathbf{w}_{t_{k+1}}|\mathbf{w}_{t_k})$ is given by the *Ornstein-Uhlenbeck* process (cf. Sec. 4.2). No memory used.

- Bayesian forgetting: Posterior approximation consists of $q_{\theta_{t_k}}(\mathbf{w})$ only. No state-space model assumption, instead uses Bayesian exponential forgetting (cf. Sec. 4.1).

- Bayesian forgetting with memory: Posterior approximation consists of $q_{\theta_{t_k}}(\mathbf{w})$ and $\mathcal{M}_t$. No state-space model assumption, instead uses Bayesian exponential forgetting (cf. Sec. 4.1).

In Tab. 2, we summarise experimental setup (hyperparameters) used for Secs. 6.1 and 6.2.

## C FACTORISATION PROPERTY OF THE GAUSSIAN VARIATIONAL APPROXIMATION

Here we derive the factorisation property of the Gaussian variational approximation distribution by expressing the natural parameters of the Gaussian approximation as a sum. This can be shown for the Gaussian approximation at a local optimum of the ELBO. For a Gaussian prior and posterior the

Table 2: Experiment setup for online experiments. $N_{t_0}$ is the number of observed samples at the first time-step and $N_{t_{1:k}}$ is the dataset size of all other time-steps. $M$ refers to the number of samples in the memory. We evaluated 5 different memory sizes for each experiment in Sec. 3 and 10 sizes for experiments in Sec. 4, where the sizes are equally spaced in the given range. $K_{\text{train}}$ and $K_{\text{term}}$ is the number of MC samples used for training and for estimating the Gaussian terms respectively. $I_{t_0}$ and $I_{t_{1:K}}$ refer to the number of iterations for the first time-step and all subsequent time-steps, respectively. The architecture denotes the number of units for each hidden layer (e.g. $[16, 16]$ denotes 2 hidden layers with 16 units each).

|  | $N_{t_0}$ | $N_{t_{1:k}}$ | $M$ | $K_{\text{train}}$ | $K_{\text{term}}$ | $I_{t_0}$ | $I_{t_{1:K}}$ | Architecture |
|---|---|---|---|---|---|---|---|---|
| Concrete | 100 | 10 | $[5\ldots25]$ | 1k | 50k | 50k | 10k | $[16, 16]$ |
| Boston | 100 | 10 | $[5\ldots25]$ | 1k | 50k | 50k | 10k | $[16, 16]$ |
| Energy | 100 | 10 | $[5\ldots25]$ | 1k | 50k | 50k | 10k | $[16, 16]$ |
| Yacht | 100 | 10 | $[5\ldots25]$ | 1k | 50k | 50k | 10k | $[16, 16]$ |
| Spam | 250 | 25 | $[10\ldots50]$ | 400 | 50k | 50k | 10k | $[32, 32]$ |
| Wine | 250 | 25 | $[10\ldots50]$ | 400 | 50k | 50k | 10k | $[32, 32]$ |
| MNIST | 2.5k | 250 | $[50\ldots250]$ | 40 | 50k | 50k | 10k | $[64, 64]$ |
| GasSensorArray | 500 | 50 | $[5\ldots50]$ | 200 | 100k | 50k | 10k | $[8, 8]$ |
| Weather | 1k | 100 | $[5\ldots50]$ | 100 | 100k | 50k | 10k | $[16, 16]$ |
| Covertype | 1k | 1k | $[10\ldots200]$ | 400 | 100k | 50k | 10k | $[32, 32]$ |

ELBO is given as

$$\mathcal{L}(\mu^*, \Sigma^*) = -\frac{1}{2}\Big(\log|\Sigma_0| - \log|\Sigma^*| - d + (\mu^* - \mu_0)^T\Sigma_0^{-1}(\mu^* - \mu_0) + Tr(\Sigma^*\Sigma_0^{-1})\Big)$$
$$+ \sum_{n=1}^{N}\mathbb{E}_{\mathbf{w}\sim q_{\theta^*}(\mathbf{w})}\Big[\log p(\mathbf{d}^{(n)}|\mathbf{w})\Big].$$

At a local optimum, we have $\frac{\partial\mathcal{L}(\mu^*, \Sigma^*)}{\partial\mu^*} = 0$, which yields

$$\sum_{n=1}^{N}\frac{\partial}{\partial\mu^*}\mathbb{E}_{\mathbf{w}\sim q_{\theta^*}(\mathbf{w})}\Big[\log p(\mathbf{d}^{(n)}|\mathbf{w})\Big] = \Sigma_0^{-1}(\mu^* - \mu_0).$$

Hence, we obtain

$$\mu^* = \mu_0 + \Sigma_0\sum_{n=1}^{N}\frac{\partial}{\partial\mu^*}\mathbb{E}_{\mathbf{w}\sim q_{\theta^*}(\mathbf{w})}\Big[\log p(\mathbf{d}^{(n)}|\mathbf{w})\Big]. \tag{14}$$

Similarly, we have $\frac{\partial\mathcal{L}(\mu^*, \Sigma^*)}{\partial\Sigma^*} = 0$, which yields

$$\sum_{n=1}^{N}\frac{\partial}{\partial\Sigma^*}\mathbb{E}_{\mathbf{w}\sim q_{\theta^*}(\mathbf{w})}\Big[\log p(\mathbf{d}^{(n)}|\mathbf{w})\Big] = -\frac{1}{2}(\Sigma^*)^{-1} + \frac{1}{2}\Sigma_0^{-1}. \tag{15}$$

Hence, we obtain

$$\Sigma^* = \Big(\Sigma_0^{-1} - 2\sum_{n=1}^{N}\frac{\partial}{\partial\Sigma^*}\mathbb{E}_{\mathbf{w}\sim q_{\theta^*}(\mathbf{w})}\Big[\log p(\mathbf{d}^{(n)}|\mathbf{w})\Big]\Big)^{-1}. \tag{16}$$

Next, we calculate the natural parameters from Eqs. (14), (16):

$$\Lambda^* = \Lambda_0 + \sum_{n=1}^{N}-2\frac{\partial}{\partial\Sigma^*}\mathbb{E}_{\mathbf{w}\sim q_{\theta^*}(\mathbf{w})}\Big[\log p(\mathbf{d}^{(n)}|\mathbf{w})\Big]$$
$$= \Lambda_0 + \sum_{n=1}^{N}\Lambda^{(n)}.$$

$$\eta^* = \Lambda^* \mu^* = \Big( \Lambda_0 + \sum_{n=1}^{N} \Lambda^{(n)} \Big) \mu^*$$

$$= \Lambda_0 \Big( \mu_0 + \Sigma_0 \sum_{n=1}^{N} \frac{\partial}{\partial \mu^*} \mathbb{E}_{\mathbf{w} \sim q_{\theta^*}(\mathbf{w})} \Big[ \log p(\mathbf{d}^{(n)}|\mathbf{w}) \Big] \Big) + \sum_{n=1}^{N} \Lambda^{(n)} \mu^*$$

$$= \Lambda_0 \mu_0 + \sum_{n=1}^{N} \Big( \frac{\partial}{\partial \mu^*} \mathbb{E}_{\mathbf{w} \sim q_{\theta^*}(\mathbf{w})} \Big[ \log p(\mathbf{d}^{(n)}|\mathbf{w}) \Big] + \Lambda^{(n)} \mu^* \Big)$$

$$= \eta_0 + \sum_{n=1}^{N} \eta^{(n)}.$$

**Monte Carlo estimation:** The natural parameters $\Lambda^{(n)}$, $\eta^{(n)}$ can be estimated with Monte Carlo, by replacing the expectation with an empirical mean. Since the parameters $\Lambda^*$ and $\Lambda_0$ (and $\eta^*$, $\eta_0$ respectively) are known, the total bias of the parameter estimates can be computed:

$$\Lambda_{\mathrm{b}} = (\Lambda^* - \Lambda_0) - \sum_{n=1}^{N} \Lambda^{(n)}, \qquad\qquad \eta_{\mathrm{b}} = (\eta^* - \eta_0) - \sum_{n=1}^{N} \eta^{(n)}.$$

We use this to reduce the bias for the individual terms:

$$\hat{\Lambda}^{(n)} = \Lambda^{(n)} - \frac{1}{N} \Lambda_{\mathrm{b}}, \qquad\qquad \hat{\eta}^{(n)} = \eta^{(n)} - \frac{1}{N} \eta_{\mathrm{b}}.$$

## D  ELBO IN RESIDUALS FORM

Here we show how the ELBO can be written in the form of Eq. (6). Let us define the variational distribution in the factorised form $q_\theta(\mathbf{w}) = Z_q^{-1} p(\mathbf{w}) \prod_{n=1}^{N} \mathbf{r}^{(n)}(\mathbf{w})$ (cf. Sec.3.1). We can then write the ELBO as

$$\mathcal{L}(\mu, \Sigma; \mathcal{D}) = \mathbb{E}_{q_\theta(\mathbf{w})} \Bigg[ \sum_{n=1}^{N} \log p(\mathbf{d}^{(n)}|\mathbf{w}) + \log p(\mathbf{w}) - \log q_\theta(\mathbf{w}) \Bigg]$$

$$= \mathbb{E}_{q_\theta(\mathbf{w})} \Bigg[ \sum_{n=1}^{N} \big( \log p(\mathbf{d}^{(n)}|\mathbf{w}) + \log \mathbf{r}^{(n)}(\mathbf{w}) - \log \mathbf{r}^{(n)}(\mathbf{w}) \big) + \log p(\mathbf{w}) - \log q_\theta(\mathbf{w}) \Bigg]$$

$$= \mathbb{E}_{q_\theta(\mathbf{w})} \Bigg[ \sum_{n=1}^{N} \big( \log p(\mathbf{d}^{(n)}|\mathbf{w}) - \log \mathbf{r}^{(n)}(\mathbf{w}) \big) + \log \frac{p(\mathbf{w}) \prod_{n=1}^{N} \mathbf{r}^{(n)}(\mathbf{w})}{q_\theta(\mathbf{w})} \Bigg]$$

$$= \mathbb{E}_{q_\theta(\mathbf{w})} \Bigg[ \sum_{n=1}^{N} \big( \log p(\mathbf{d}^{(n)}|\mathbf{w}) - \log \mathbf{r}^{(n)}(\mathbf{w}) \big) + \log \frac{Z_q \cdot q_\theta(\mathbf{w})}{q_\theta(\mathbf{w})} \Bigg]$$

$$= \log Z_q + \sum_{n} \mathbb{E}_{q_\theta(\mathbf{w})} \big[ \log p(\mathbf{d}^{(n)}|\mathbf{w}) - \log \mathbf{r}^{(n)}(\mathbf{w}) \big].$$

## E  MEMORY UPDATE SCORE FUNCTION

In Eq. 8, the expectation involving Gaussian terms can be calculated analytically:

$$\mathbb{E}_{\tilde{q}_{\theta_{t_k}}(\mathbf{w})} \Big[ \log \mathbf{r}_{t_k}^{(m)}(\mathbf{w}) \Big] = \int \tilde{q}_{\theta_{t_k}}(\mathbf{w}) \big( \eta^{(n)} \mathbf{w} - \frac{1}{2} \Lambda^{(n)} \mathbf{w}^2 \big) d\mathbf{w}$$

$$= \eta^{(n)} \mu^{(n)} - \frac{1}{2} \Lambda^{(n)} \big( (\mu^*)^2 + \Sigma^* \big)$$

$$= \eta^{(n)} (\Lambda^*)^{-1} \eta^* - \frac{1}{2} \Lambda^{(n)} \big( (\Lambda^*)^{-1} \eta^* \big)^2 - \frac{1}{2} \Lambda^{(n)} (\Lambda^*)^{-1}$$

The expectation involving non-Gaussian terms (in Eq. 8) has no closed-form solution. We therefore estimate $\mathbb{E}_{\tilde{q}_{\theta_{t_k}}(\mathbf{w})} \big[ \log p(\mathbf{d}_{t_k}^{(n)}|\mathbf{w}) \big]$ using Monte-Carlo.

## F  BAYESIAN FORGETTING - RECURSIVE FORMULATION

Here we show how Bayesian forgetting can be rearranged into a recursive formulation. We first bring this formula into a similar form as Eq. (1), extracting the most recent likelihood term:

$$p(\mathbf{w}|\mathcal{D}_{t_1:t_{K+1}}) \propto p_0(\mathbf{w}) \cdot \prod_{k=1}^{K+1} p(\mathcal{D}_{t_k}|\mathbf{w})^{(1-\epsilon)^{\frac{t_{K+1}-t_k}{\tau}}}$$

$$= p_0(\mathbf{w}) \cdot \prod_{k=1}^{K} p(\mathcal{D}_{t_k}|\mathbf{w})^{(1-\epsilon)^{\frac{t_{K+1}-t_k}{\tau}}} \cdot p(\mathcal{D}_{t_{K+1}}|\mathbf{w}).$$

The first two terms can be rewritten as

$$p_0(\mathbf{w}) \cdot \prod_{k=1}^{K} p(\mathcal{D}_{t_k}|\mathbf{w})^{(1-\epsilon)^{\frac{t_{K+1}-t_k}{\tau}}} = p_0(\mathbf{w}) \cdot \prod_{k=1}^{K} p(\mathcal{D}_{t_k}|\mathbf{w})^{(1-\epsilon)^{\frac{t_{K+1}-t_K+t_K-t_k}{\tau}}}$$

$$= p_0(\mathbf{w}) \cdot \prod_{k=1}^{K} p(\mathcal{D}_{t_k}|\mathbf{w})^{(1-\epsilon)^{\frac{t_K-t_k}{\tau}} \cdot (1-\epsilon)^{\frac{t_{K+1}-t_K}{\tau}}}$$

$$= p_0(\mathbf{w}) \cdot \left( \prod_{k=1}^{K} p(\mathcal{D}_{t_k}|\mathbf{w})^{(1-\epsilon)^{\frac{t_K-t_k}{\tau}}} \right)^{(1-\epsilon)^{\frac{t_{K+1}-t_K}{\tau}}}$$

$$\propto p_0(\mathbf{w}) \cdot \left( \frac{p(\mathbf{w}|\mathcal{D}_{t_1:t_K})}{p_0(\mathbf{w})} \right)^{(1-\epsilon)^{\frac{t_{K+1}-t_K}{\tau}}}$$

$$= p_0(\mathbf{w})^{1-(1-\epsilon)^{\frac{t_{K+1}-t_K}{\tau}}} \cdot p(\mathbf{w}|\mathcal{D}_{t_1:t_K})^{(1-\epsilon)^{\frac{t_{K+1}-t_K}{\tau}}}$$

Hence, we have shown that the posterior can be expressed recursively as

$$p(\mathbf{w}|\mathcal{D}_{t_1:t_{k+1}}) \propto p_0(\mathbf{w})^{1-(1-\epsilon)^{\Delta t_{k+1}/\tau}} p(\mathbf{w}|\mathcal{D}_{t_1:t_k})^{(1-\epsilon)^{\Delta t_{k+1}/\tau}} p(\mathcal{D}_{t_{k+1}}|\mathbf{w}).$$

The parameters of the Gaussian part $q_{\theta_{t_k}}(\mathbf{w})$ of the posterior approximation (after applying the forgetting operation) can be calculated easily from the above equation.

**Natural parameters:**

$$\Lambda_{t_{k+1}} = \Lambda_0 \cdot (1 - (1-\epsilon)^{\frac{t_{k+1}-t_k}{\tau}}) + \Lambda_{t_k} \cdot ((1-\epsilon)^{\frac{t_{k+1}-t_k}{\tau}})$$

$$\eta_{t_{k+1}} = \eta_0 \cdot (1 - (1-\epsilon)^{\frac{t_{k+1}-t_k}{\tau}}) + \eta_{t_k} \cdot ((1-\epsilon)^{\frac{t_{k+1}-t_k}{\tau}})$$

**Covariance parameter:**

$$\Sigma_{t_{k+1}} = \left( \Lambda_0 \cdot (1 - (1-\epsilon)^{\frac{t_{k+1}-t_k}{\tau}}) + \Lambda_{t_k} \cdot ((1-\epsilon)^{\frac{t_{k+1}-t_k}{\tau}}) \right)^{-1}$$

$$= \left( \Sigma_0^{-1} \cdot (1 - (1-\epsilon)^{\frac{t_{k+1}-t_k}{\tau}}) + \Sigma_{t_k}^{-1} \cdot ((1-\epsilon)^{\frac{t_{k+1}-t_k}{\tau}}) \right)^{-1}$$

**Location parameter:**

$$\mu_{t_{k+1}} = \Sigma_{t_{k+1}} \cdot \eta_{t_{k+1}}$$

$$= \Sigma_{t_{k+1}} \left( \eta_0 \cdot (1 - (1-\epsilon)^{\frac{t_{k+1}-t_k}{\tau}}) + \eta_{t_k} \cdot ((1-\epsilon)^{\frac{t_{k+1}-t_k}{\tau}}) \right)$$

$$= \Sigma_{t_{k+1}} \left( \Sigma_0^{-1} \mu_0 \cdot (1 - (1-\epsilon)^{\frac{t_{k+1}-t_k}{\tau}}) + \Sigma_{t_k}^{-1} \mu_{t_k} \cdot ((1-\epsilon)^{\frac{t_{k+1}-t_k}{\tau}}) \right)$$

## G  PSEUDO-ALGORITHM

We provide the pseudo algorithm of GRS (Sec. 3) with Bayesian forgetting in Alg. 1. The computational complexity (at each of the $K$ time-steps) is dominated by i) the minimisation of the KL

divergence, ii) estimating the Gaussian factors, and iii) scoring the memory. The KL minimisation requires $I_{t_k}$ sequential iterations with $N_{t_k} + M$ data samples and $K_{\text{train}}$ Monte Carlo samples. The latter can both be processed in parallel on parallel hardware. The estimation of the Gaussian factors requires $N_{t_k} + M$ sequential iterations with $K_{\text{term}}$ parallel Monte Carlo samples. The dominating computation for calculating the scores is the evaluation of the likelihood for $N_{t_k} + M$ data samples and $K_{\text{train}}$ Monte Carlo samples, both of which can be processed in parallel. The highest scoring candidate memory is given by the top-$M$ highest scoring data points, thus, the computational complexity of Eq. (8) is only linear in the number of samples.

---

**Algorithm 1** Gaussian Residual Scoring with Bayesian forgetting. The function *EstimateGaussianFactors* corresponds to Eqs. (5a), (5b) (cf. also App. C). The function *ApplyForgetting* corresponds to Eq. (12). Note that $p_{t_k}(\mathbf{w})$ includes the adapted likelihood of the memory and all subsequent functions involving the memory use this adapted likelihood.

---

Inputs: $p_0(\mathbf{w})$, $q_{\theta_{t_0}}(\mathbf{w})$, $\tau$, $K$
**for** $k$ in $[0...K]$ **do**
   $t_k = \text{GetTimeStamp}(k)$
   $\Delta t_k = t_k - t_{k-1}$
   $\mathcal{D}_{t_k} = \text{GetData}(t_k)$
   **if** $k == 0$ **then**
      $p_{t_k}(\mathbf{w}) = p_0(\mathbf{w})$
   **else**
      $p_{t_k}(\mathbf{w}) = \text{ApplyForgetting}\big(p_0(\mathbf{w}), q_{\theta_{t_{k-1}}}, \mathcal{M}_{t_{k-1}}, \Delta t_k\big)$ {Sec. 4.1}
   **end if**
   $\tilde{q}_{\theta_{t_k}}(\mathbf{w}) = \text{argmin}_{q_\theta}\, \text{KL}\big[q_\theta(\mathbf{w}) \parallel \tilde{Z}_{t_k}^{-1} p_{t_k}(\mathbf{w})\, p(\mathcal{D}_{t_k}|\mathbf{w})\big]$ {Sec. 3.2, Sec. 4.1}
   $\{\mathbf{r}_{t_k}(\mathbf{w}; \mathbf{d}_{t_k})\}_{\mathbf{d}_{t_k} \in \mathcal{D}_{t_k} \cup \mathcal{M}_{t_{k-1}}} = \text{EstimateGaussianFactors}\big(\tilde{q}_{\theta_{t_k}}(\mathbf{w}), \mathcal{D}_{t_k}, \mathcal{M}_{t_{k-1}}\big)$ {Sec. 3.1}
   $\mathcal{M}_{t_k} = \text{argmax}_{\mathcal{M}}\, S_{t_k}(\mathcal{M}; \mathcal{D}_{t_k} \cup \mathcal{M}_{t_{k-1}})$ {Sec. 3.2}
   **if** $|\mathcal{D}_{t_k}| \leq |\mathcal{M}_{t_k}|$ **then**
      $q_{\theta_{t_k}}(\mathbf{w}) = p_{t_k}(\mathbf{w}) \prod_{\mathbf{d}_{t_k} \notin \mathcal{M}_{t_k}} \mathbf{r}_{t_k}(\mathbf{w}; \mathbf{d}_{t_k})$ {Sec. 3.3}
   **else**
      $q_{\theta_{t_k}}(\mathbf{w}) = \tilde{q}_{\theta_{t_k}}(\mathbf{w}) \,/ \prod_{\mathbf{d}_{t_k} \in \mathcal{M}_{t_k}} \mathbf{r}_{t_k}(\mathbf{w}; \mathbf{d}_{t_k})$ {Sec. 3.3}
   **end if**
**end for**

---

## H PRIOR PARAMETERS

Here we develop a heuristic to choose the initial prior $p_0(\mathbf{w})$. As this will not be specific to the online or continual setting, we drop the time index in this section, denoting the prior as $p(\mathbf{w})$. Furthermore, we consider only Gaussian distributions with a diagonal covariance matrix. Assume that the data is standardised, that is, the first two moments are zero and one. A reasonable choice for the prior parameters is such that the first two moments of the prior-predictive distribution equals the first two moments of the data distribution. We go one step further and constrain the pre-activations of every neural network layer to have moments zero and one. Denote all weight matrices and weight biases by $\mathbf{w} = \{\mathbf{W}_l\}_l \cup \{\mathbf{b}_l\}_l$, and let $\mathbf{x}_0$ denote the input data. Let us further denote the pre-activation (before non-linearity) of layer $l$ and unit $i$ as follows.

$$\mathbf{x}_l^i = \sum_j \mathbf{W}_l^{i,j} f_{l-1}(\mathbf{x}_{l-1}^j) + \mathbf{b}_l^i.$$

The constraints are then given as follows.

$$\mathbb{E}_{\mathbf{w} \sim p(\mathbf{w})}\Big[\mathbb{E}_{\mathbf{x}_0 \sim p(\mathbb{D})}\big[\mathbf{x}_l^i\big]\Big] = 0, \qquad\qquad \mathbb{E}_{\mathbf{w} \sim p(\mathbf{w})}\Big[\mathbb{E}_{\mathbf{x}_0 \sim p(\mathbb{D})}\big[(\mathbf{x}_l^i)^2\big]\Big] = 1.$$

The first constraint can be easily fulfilled by setting the prior mean to zero for all parameters.

$$\mu_l^{i,j} = 0.$$

This follows immediately from $\mathbf{W}_l \perp\!\!\!\perp f_{l-1}(\mathbf{x}_{l-1})$ and the expectation of products of independent random variables. The second moment can then be calculated as follows.

$$
\begin{aligned}
\mathbb{E}_{\mathbf{w}\sim p(\mathbf{w}), \mathbf{x}_0 \sim p(\mathbb{D})}\left[(\mathbf{x}_l^i)^2\right] &= \mathbb{E}_{\mathbf{w}\sim p(\mathbf{w})}\left[\mathbb{E}_{\mathbf{x}_0 \sim p(\mathbb{D})}\left[(\mathbf{x}_l^i)^2\right]\right] \\
&= \mathbb{E}_{\mathbf{w}\sim p(\mathbf{w})}\left[\mathrm{Var}_{\mathbf{x}_0 \sim p(\mathbb{D})}\left[\mathbf{x}_l^i\right] + 0\right] \\
&= \mathbb{E}_{\mathbf{w}\sim p(\mathbf{w})}\left[\mathrm{Var}_{\mathbf{x}_0 \sim p(\mathbb{D})}\left[\sum_j^{N_{l-1}} \mathbf{W}_l^{i,j} f_{l-1}(\mathbf{x}_{l-1}^j) + \mathbf{b}_l^i\right]\right] \\
&= \mathbb{E}_{\mathbf{w}\sim p(\mathbf{w})}\left[\sum_j^{N_{l-1}} \left(\mathbf{W}_l^{i,j}\right)^2 \cdot \mathrm{Var}_{\mathbf{x}_0 \sim p(\mathbb{D})}\left[f_{l-1}(\mathbf{x}_{l-1}^j)\right]\right] \\
&= \sum_j^{N_{l-1}} \mathbb{E}_{\mathbf{w}\sim p(\mathbf{w})}\left[\left(\mathbf{W}_l^{i,j}\right)^2 \cdot \mathrm{Var}_{\mathbf{x}_0 \sim p(\mathbb{D})}\left[f_{l-1}(\mathbf{x}_{l-1}^j)\right]\right] \\
&= \sum_j^{N_{l-1}} \mathbb{E}_{\mathbf{w}\sim p(\mathbf{w})}\left[\left(\mathbf{W}_l^{i,j}\right)^2\right] \cdot \mathbb{E}_{\mathbf{w}\sim p(\mathbf{w})}\left[\mathrm{Var}_{\mathbf{x}_0 \sim p(\mathbb{D})}\left[f_{l-1}(\mathbf{x}_{l-1}^j)\right]\right] \\
&=: \sum_j^{N_{l-1}} \mathbb{E}_{\mathbf{w}\sim p(\mathbf{w})}\left[\left(\mathbf{W}_l^{i,j}\right)^2\right] \cdot c_{f_{l-1}} \\
&= c_{f_{l-1}} \cdot \sum_j^{N_{l-1}} \mathrm{Var}_{\mathbf{w}\sim p(\mathbf{w})}\left[\mathbf{W}_l^{i,j}\right] \\
&= N_{l-1} \cdot c_{f_{l-1}} \cdot \mathrm{Var}_{\mathbf{w}\sim p(\mathbf{w})}\left[\mathbf{W}_l^{i,j}\right].
\end{aligned}
$$

Here we introduced $c_{f_{l-1}}$ to denote a correction factor for the non-linearity $f_{l-1}$. In case of the linear function, we will have $c_{f_{l-1}} = 1$. For arbitrary non-linearities, we can estimate this factor numerically, assuming that the pre-activations are distributed according to $\mathcal{N}(0, 1)$.

$$
c_{f_{l-1}} = \mathrm{Var}_{\mathbf{x}_{l-1}^j \sim \mathcal{N}(0,1)}\left[f_{l-1}(\mathbf{x}_{l-1}^j)\right]
$$

This can be done beforehand and the factors for common activation functions can be stored in a lookup table. Finally, plugging in the constraint for the second moment in the above equation, we obtain the following prior variance.

$$
\left(\sigma_l^{i,j}\right)^2 = \mathrm{Var}_{\mathbf{w}\sim p(\mathbf{w})}\left[\mathbf{W}_l^{i,j}\right] = \frac{1}{N_{l-1} c_{f_{l-1}}} \tag{17}
$$

## I  POSTERIOR INITIALISATION

It is known that a proper initialisation of standard neural networks is crucial for the optimisation process Glorot & Yoshua Bengio (2010); He et al. (2015). In Bayesian neural networks, the matter becomes even more complicated, since we have to deal additionally with the variance of the Monte Carlo estimator due to re-parametrisation. Analogous to the choice of prior parameters, we seek a posterior initialisation that yields the first two moments of zero and one. A naive attempt would be to initialise the posterior with the prior parameters. However, the significant noise in the Monte Carlo estimation typically leads to bad optimisation results and even numerical instabilities. We propose an initialisation method which fulfills our constraints but allows us determine the variance of the initial posterior with two hyperparameters $\alpha$ and $\beta$.

Let us denote the mean and log-scale parameters of the approximate posterior as $\theta = \{\theta_\mu, \theta_{\log \sigma}\}$. We choose the following initialisation distributions.

$$
q(\mathbf{w}) = \mathcal{N}(\mathbf{w}; \theta_\mu,\ e^{2\theta_{\log \sigma}}),
$$

where

$$
p(\theta_\mu) = \mathcal{N}\left(\theta_\mu; \mu_{\theta_\mu}, \sigma_{\theta_\mu}^2\right),
$$

and

$$p(\theta_{\log\sigma}) = \mathcal{N}\big(\theta_{\log\sigma}; \mu_{\theta_{\log\sigma}}, \sigma^2_{\theta_{\log\sigma}}\big).$$

Here and in the following, we dropped the time index for the approximate posterior, as well as the indices $l$, $i$, and $j$ for the model parameters $\theta$.

We follow a similar derivation as in Sec. H. As for the prior, the mean of the initialisation distribution will be zero for all parameters.

$$\mu_{\theta_\mu} = 0.$$

For the second moment, the derivation is as follows.

$$\mathbb{E}_{\theta\sim p(\theta), \mathbf{w}\sim q(\mathbf{w}|\theta), \mathbf{x}_0\sim p(\mathbb{D})}\Big[(\mathbf{x}_l^i)^2\Big] = \mathbb{E}_{\theta\sim p(\theta)}\bigg[\mathbb{E}_{\mathbf{w}\sim q(\mathbf{w}|\theta)}\Big[\mathbb{E}_{\mathbf{x}_0\sim p(\mathbb{D})}\Big[(\mathbf{x}_l^i)^2\Big]\Big]\bigg]$$

$$= \mathbb{E}_{\theta\sim p(\theta)}\Bigg[\mathbb{E}_{\mathbf{w}\sim q(\mathbf{w}|\theta)}\bigg[\mathrm{Var}_{\mathbf{x}_0\sim p(\mathbb{D})}\Big[\sum_j^{N_{l-1}} \mathbf{W}_l^{i,j} \cdot f_{l-1}(\mathbf{x}_{l-1}^j)\Big] + 0\bigg]\Bigg]$$

$$= \mathbb{E}_{\theta\sim p(\theta)}\Bigg[\mathbb{E}_{\mathbf{w}\sim q(\mathbf{w}|\theta)}\bigg[\sum_j^{N_{l-1}} (\mathbf{W}_l^{i,j})^2 \mathrm{Var}_{\mathbf{x}_0\sim p(\mathbb{D})}\Big[\mathbf{x}_{l-1}^j\Big]\bigg]\Bigg]$$

$$= \mathbb{E}_{\theta\sim p(\theta)}\Bigg[\mathbb{E}_{\mathbf{w}\sim q(\mathbf{w}|\theta)}\bigg[\sum_j^{N_{l-1}} (\mathbf{W}_l^{i,j})^2\bigg]\Bigg] \cdot \mathbb{E}_{\theta\sim p(\theta)}\Bigg[\mathbb{E}_{\mathbf{w}\sim q(\mathbf{w}|\theta)}\bigg[\mathrm{Var}_{\mathbf{x}_0\sim p(\mathbb{D})}\Big[f_{l-1}(\mathbf{x}_{l-1}^j)\Big]\bigg]\Bigg]$$

$$= \sum_j^{N_{l-1}} \mathbb{E}_{\theta\sim p(\theta)}\bigg[\mathbb{E}_{\mathbf{w}\sim q(\mathbf{w}|\theta)}\Big[(\mathbf{W}_l^{i,j})^2\Big]\bigg] \cdot \mathbb{E}_{\theta\sim p(\theta)}\Bigg[\mathbb{E}_{\mathbf{w}\sim q(\mathbf{w}|\theta)}\bigg[\mathrm{Var}_{\mathbf{x}_0\sim p(\mathbb{D})}\Big[f_{l-1}(\mathbf{x}_{l-1}^j)\Big]\bigg]\Bigg]$$

$$=: \sum_j^{N_{l-1}} \mathbb{E}_{\theta\sim p(\theta)}\Big[\theta_\mu^2 + e^{2\cdot\theta_{\log\sigma}}\Big] \cdot c_{f_{l-1}}$$

$$= N_{l-1} \cdot c_{f_{l-1}} \cdot \mathbb{E}_{\theta\sim p(\theta)}\Big[\theta_\mu^2 + e^{2\cdot\theta_{\log\sigma}}\Big]$$

$$= N_{l-1} \cdot c_{f_{l-1}} \cdot \Big(\mathbb{E}_{\theta\sim p(\theta)}\Big[\theta_\mu^2\Big] + \mathbb{E}_{\theta\sim p(\theta)}\Big[e^{2\cdot\theta_{\log\sigma}}\Big]\Big)$$

$$= N_{l-1} \cdot c_{f_{l-1}} \cdot \Big(\mathbb{E}_{\theta\sim p(\theta)}\big[\theta_\mu\big]^2 + \mathrm{Var}_{\theta\sim p(\theta)}\big[\theta_\mu\big] + e^{2\mathbb{E}[\theta_{\log\sigma}] + 2\mathrm{Var}[\theta_{\log\sigma}]}\Big)$$

$$= N_{l-1} \cdot c_{f_{l-1}} \cdot \Big(\mu_{\theta_\mu}^2 + \sigma_{\theta_\mu}^2 + e^{2\mu_{\theta_{\log\sigma}} + 2\sigma_{\theta_{\log\sigma}}^2}\Big)$$

$$= N_{l-1} \cdot c_{f_{l-1}} \cdot \Big(\sigma_{\theta_\mu}^2 + e^{2\mu_{\theta_{\log\sigma}} + 2\sigma_{\theta_{\log\sigma}}^2}\Big).$$

Hence, the second constraint is as follows.

$$\frac{1}{N_{l-1} \cdot c_{f_{l-1}}} = \sigma_{\theta_\mu}^2 + e^{2\mu_{\theta_{\log\sigma}} + 2\sigma_{\theta_{\log\sigma}}^2}.$$

In contrast to Sec. H, we are now under-constrained by 2 parameters. We therefore introduce two hyperparameters $\alpha$ and $\beta$. We first determine $\alpha := \sigma_{\theta_{\log\sigma}}$, for which we generally choose small values $\alpha \approx 0$ ($\alpha = 0$ corresponds to initialising all posterior variances in the given layer with the same value). The second hyperparameter $\beta \in [0, 1]$ determines how much of the total variance is due to the variance of the location parameter and how much variance is due to the variance of the log-scale parameter. Inserting $\alpha$ and introducing $\beta$ we obtain the following equations.

$$\sigma_{\theta_\mu}^2 = \frac{\beta}{N_{l-1} \cdot c_{f_{l-1}}},$$

and

$$e^{2\mu_{\theta_{\log\sigma}} + 2\alpha^2} = \frac{1 - \beta}{N_{l-1} \cdot c_{f_{l-1}}}.$$

Solving the last equation for $\mu_{\theta_{\log \sigma}}$, the result is as follows.

$$\mu_{\theta_{\log \sigma}} = \frac{1}{2} \log \frac{(1-\beta) \cdot e^{-2\alpha^2}}{N_{l-1} \cdot c_{f_{l-1}}}$$

We choose $\alpha = 0.001$ and $\beta = 0.999$ in all experiments.

**A note on the relation to initialisation methods for deterministic neural networks.** Our result is similar to the initialisation methods from Glorot & Yoshua Bengio (2010) and He et al. (2015). The difference is in the correction factor $c_{f_{l-1}}$. Whereas Glorot & Yoshua Bengio (2010) considers linear functions (or tanh in the linear regime), both methods base their derivation on the assumption that every data sample $\mathbf{x}_0$ is processed by a different, random neural network with independent weights, drawn from the initialisation distribution. The assumption is made explicit in (He et al., 2015) by the use of the variance of products of independent variables rule. We note that this assumption is false for both the initialisation of deterministic neural networks, as well as the graphical model assumption in Bayesian neural networks. Consequently, (He et al., 2015) obtains different correction factors (in their case for relu and leaky relu), taking into account the mean after the forward-pass through the non-linearity.

