# OpenReview forum: "Continual Learning with Bayesian Neural Networks for Non-Stationary Data"
_ICLR.cc/2020/Conference — Accept (Poster)_

### Official Review · AnonReviewer2 · 2019-10-22
**Official Blind Review #2**

**Rating:** 6

**Review:**

Post-rebuttal: My questions below have been addressed and the submission has been modified accordingly. My concern regarding novelty remains unchanged but I still suggest acceptance since the contributions are of practical interests and the paper is well written.

1. Summary:
This proposes considers neural networks training with non-stationary streaming data. To address online inference, the paper uses variational online updates and a running memory (coreset) summarising the data points seen so far, as recently used in the variational continual learning approach. First, the paper identifies a gap in this recent approach that coreset point selection process does not take into account the approximation quality. The paper develops a sound procedure to select the memory points, essentially to identify “difficult” data points under the current posterior and put them into the running memory. Second, to handle “concept drift”, the paper considers imposing some decaying on the likelihood of past data and derives post-hoc variational posterior updates for this case. Two contributions are validated on some streaming regression and classification tasks.

2. Opinion and rationales

I’m leaning towards “accept” for this paper since it presents two interesting contributions (albeit of incremental novelty)  to the approximate inference area, it has clear execution and super clean presentation, and the experiments clearly demonstrate the values of the proposed approaches.

I would like the paper to clarify the following:

a. I find the objective for selecting memory points interesting, but am wondering if using individual predictability terms by themselves are good enough to select these points. Perhaps, after some thoughts, memory is perhaps not the right word to characterise these points.

It seems to me (i) these terms indicate how well the current posterior predicts the data points and thus this objective will tend to favour points with low predictive likelihood to be selected. That is, these points are important when *all other points* are presented, (ii) there will be *no diversity* in the memory as illustrated in the 2d classification example.

This means the points selected here have quite different characteristics compared to coreset points or the full dataset in general. Coreset points ideally can compactly represent the full dataset and can be used for inference in place of the full dataset.

That said, the contribution presented here is very useful. Just that I’m not sure how well this will work in more challenging continual learning set-up where diversity is important for a long sequence of diverse tasks.

b. The paper presents a post-hoc modification to approximate posterior (‘s mean and variance) to account for the decay in the likelihood contribution from past data. I wonder if this post-hoc adjustment is identical to running approximate inference with the adjusted likelihood?

c. the adaptation parameters need to be tuned/known in advance, which seems to be a disadvantage of the current approach. Is the update presented here somewhat robust to mis-specification of these parameters? How would these be tuned in a more practical setting in which data arrive sequentially and we might not know the underlying “concept”.

3. Minor details:

Proxys -> proxies


**Experience Assessment:**

I have published one or two papers in this area.

**Review Assessment: Checking Correctness Of Derivations And Theory:**

I carefully checked the derivations and theory.

**Review Assessment: Checking Correctness Of Experiments:**

I assessed the sensibility of the experiments.

**Review Assessment: Thoroughness In Paper Reading:**

I read the paper thoroughly.

---

> ### Author Response · Authors · 2019-11-14
> **Response to reviewer questions**
>
> We thank the reviewer for the kind and helpful review. We address all questions in the same order as asked (ans split question a) in two parts).
>
>
> - a1) Scoring the predictability: Note that the proposed score does not only involve individual predictability of data samples, that is, the score is not just the expected log-likelihood. Eq. (8) shows that the score is the (expectation of) difference terms between log-likelihood and log-approximate-likelihood. The difference is important: The score accounts for the change in the ELBO (cf. Eq. (6)) if the data not included in the memory is replaced by its Gaussian approximation. It is correct that our score tends to select data that can not be well predicted (Fig. 3d). However, we should emphasise that this is an empirical observation and not maximised by the score. For example, data close to the decision boundary also tends to be preferred to mis-labeled data. If the likelihood function is close to its Gaussian approximation (under the approximate posterior expectation), then the corresponding data will have a low score even if the data has low predictability/expected log-likelihood. We emphasised this in the revision of the paper.
>
> - a2) non-diversity: The reviewer is correct in that our score function does not directly account for diversity. This indeed makes our memory conceptually different from a coreset (as e.g. in VCL). We do not want to obtain a compressed dataset that summarises all previous data and is hence diverse. We seek a running memory that is complementary to the Gaussian distribution in the sense that it improves the overall posterior approximation of Eq. 3. Diversity is not a by itself a goal to achieve this, but it is likely that the memory accumulates diverse data over time. We visualised the running memory of MNIST digits at multiple time-steps; the plots are added to the supplementary material (Fig. 9) in the revision of the paper. As can be seen, the memory tends to select non-typical data. At the same time, the digits are also quite diverse, including samples from all classes.
>
> - b) This is a interesting question. Unfortunately, applying BF (posthoc) to the posterior approximation is not identical to approximating the posterior with a likelihood adjusted by BF (adhoc). If we would be able to do exact sequential inference with BF, then the results would be identical. However, the projections we have to apply at every step generally lead to different results. In other words, while Eq. (11) is an exact recursive form of Eq. (10)---which can be considered as the offline variant---the online VB approximation of Eq. (11) is not identical to the (offline) VB approximation of Eq. (10).
>
> - c) Yes, indeed the adaptation parameters are hyperparameters that need to be optimised. For this reason, we performed a grid/line search (cf. Fig. (6)), showing the dependency on the hyperparameters. On the other hand, for both methods, there is a practical interpretation of these parameters. i) For BF, 1/epsilon is the effective sample size. ii) For the OU process, 1/theta is the half-life of the mean parameter. We can calculate the time until the current state is faded out to a desired percentage using the equations for the moments (OU) and the canonical parameters (BF) provided in the paper.

---

> > ### Comment · AnonReviewer2 · 2019-11-14
> > **Thanks for the response**
> >
> > Thank you for the thoughtful reply. It would be great if these clarifications could be included in the next version of the paper.

---

> > > ### Author Response · Authors · 2019-11-15
> > > **Added clarifications to paper; pointers**
> > >
> > > Thank you for the fast reply. Question a) was actually already included in the last revision of the paper. However, in the latest revision, we expanded on a1); we now point to the plots and discussion regarding a2) in the main text; and we added the clarifications for b) and c) as well.
> > >
> > > Here are the pointers to the relevant paragraphs:
> > > - a1) Sec. 6.1, 1st paragraph (page 9): "We can make the empirical observation ...".
> > > - a2) Sec. 6.1, 1st paragraph (page 9), the last sentence points to Fig. 9, App. A.1 (page 15). Sec A.1 also includes a short discussion about the diversity.
> > > - b) Sec. 4.1, last paragraph (page 6): "We note, however, that while Eq. (11) is..."
> > > - c) Sec. 4.1, last paragraph (page 6): "For tuning the hyper parameter ...";
> > > Sec. 4.2, last paragraph (page 7): "The hyperparameter ..."
> > >
> > > Thank you again, adding these clarifications indeed improved the clarity paper.

---

### Official Review · AnonReviewer3 · 2019-10-24
**Official Blind Review #3**

**Rating:** 6

**Review:**

Contributions:
The paper extends variational continual learning with memory. In this setting the posterior distribution of the model parameters is approximated using a mean-field Gaussian approximation and a small set of datapoints is kept in a memory to combat catastrophic forgetting.

- The paper proposes a new rule for updating the memory and the Gaussian approximation after examining each batch of points. For the memory update, instead of the previously used k-means or random sampling, it examines the factor 'r(d|w)' that each datapoint contributes and selects the most significant ones.

- A second contribution is the use of two adaptation methods in the online learning setting in case there is a distribution shift in the data: Bayesian forgetting and the Ornstein-Uhlenbeck process.

(1) The first contribution, the new memory update rule and the Gaussian update, is novel to my knowledge. The idea is to calculate the Gaussian factor that each datapoint contributes to the posterior (the update corresponding to each datapoint in the message-passing interpretation) and the select the points that contribute the most. Effectively, it calculates the change in the ELBO if each candidate datapoint was moved to the memory and selects the set that minimizes the ELBO of the remaining points. This  is a sensible heuristic because we want to keep datapoints in the memory that contribute the most to the posterior approximation. If a datapoint contributes nothing to the posterior then it shouldn't be kept in memory.

The idea clever and it seems to work well in practice. Questions:
- The Gaussian that each datapoint contributes is an approximation in the sense that if we take that datapoint and its contribution away, the remaining approximate posterior might be suboptimal. This has the consequence that S_tk (eq. 8) is only an approximation to what the optimal ELBO would be without the datapoints selected for the memory. Did the authors conduct experiments or have thoughts on how well S_tk approximates the ELBO of the variational approximation of the remaining points?

-Regarding the experiments, there is a comparison to k-centre and random selection, both of which are proposed in VCL. I am very surprised to see in the experiments that both of these are outperformed by 'no-memory'. For example on 'energy', the baselines at t50 are outperformed by the non memory model at t0. Can this all be contributed to the baselines training their variational approximation without the memory at t0?

- It is unclear whether the gains can be contributed to the new Gaussian update rule or the new memory update rule. To see the contribution of each, there should be an experiment where the proposed Gaussian update rule is used along with k-centre for the memory update. The fact the the no-memory model performed to close to GRS suggests that k-centre with the new Gaussian update rule would be even closer to it.

(2) The second contribution of the paper is the use of Bayesian forgetting and OU to use to deal with the data distribution shift. It shows how these two approaches can be used along with the proposed Gaussian and memory updates. The paper is already very long, so there likely isn't any space for fleshing out this section, but it would be nice to have the experimental results included in the main paper, because most of the experiments are left to the appendix. Perhaps it would be a good idea to focus only on Bayesian forgetting (or the OU process) and try to shorten the section a bit.

In terms of writing and clarity, I have no complaints. The paper is well written and easy to understand.

Minor:
- 3.2 'If the likelihood term p(dtk|w) is well approximated by r(w;dtk)' -  I find this sentence a bit confusing. What does it mean for p(dtk|w) to be well approximated by  r(w;dtk)?
- 3.3 'In order to reduce the variance of the Monte-Carlo approximation'. What is the MC approximation made here?
- The method never actually uses the assumption that the datapoints are sampled i.i.d.. The algorithm should still work if the datapoints are not examined in a random order, e.g. consider seeing all the images with label '0' and then all the images with label '1' etc.. This would likely degrade the performance significantly but the method should still work.

Overall assessment:
Pros: I like the ideas in the paper and they are presented well.
Cons: The paper is a bit too long. The experiments could more thoroughly investigate the source of the gains.

**Experience Assessment:**

I have read many papers in this area.

**Review Assessment: Checking Correctness Of Derivations And Theory:**

I assessed the sensibility of the derivations and theory.

**Review Assessment: Checking Correctness Of Experiments:**

I assessed the sensibility of the experiments.

**Review Assessment: Thoroughness In Paper Reading:**

I read the paper at least twice and used my best judgement in assessing the paper.

---

> ### Author Response · Authors · 2019-11-14
> **Response to reviewer questions**
>
> We thank the reviewer for the detailed review. We will address all questions and suggestions in the same order.
>
> Regarding the major points:
> - The role of Stk is not to score the ELBO of the variational approximation of the remaining points (not in memory). Instead, Stk considers the whole posterior approximation (memory + Gaussian), it scores which p(d|w) can be approximated by r(w;d) (and thus by q) and which data should be kept in memory. The data corresponding to factors r(w; d) is never completely removed, we use  r(w; d)  to replace the corresponding p(d|w). However, the reviewer is right in that the Gaussian variational distribution obtained by (re-)fitting with Dtk U Mtk-1 \ Mtk is not the same as the Gaussian distribution obtained by removing the contributions through the terms r(w; d). While refitting is computationally more expensive, we also conducted experiments which show that it leads to worse results (cf. Fig. 8 in the supplementary material).
>
> - The strong performance of "no memory" and the weak performance of the "k-center" and "random" baselines are indeed surprising. As the reviewer correctly observed, it is indeed the case that the bad performance can be mostly attributed to the baselines training the variational approximation without memory. This made a significant difference on the smaller datasets (Concrete, Energy, Boston, Yacht). We attribute this problem to the fact that variational Bayesian Neural Networks (BNN) perform very poor on small datasets due to overregularisation (cf. Fig 1).
> Although we spent significant effort trying to solve this issue, we were only able to reduce it. This was however not the scope of this paper. We would like to share some insights: Our empirical results suggest that the overregularisation is not dominated by optimisation problems or a mis-sepcified prior: i) Reducing the gradient variance (e.g. using a very large amount of MC samples, local reparametrisation, momentum-based optimisers) and using KL annealing or constrained optimisation can reduce but not solve this problem. ii) Changing the prior variance gave mixed, inconclusive results. Note that not only a too small but also a too large prior variance can lead to over-regularisation in Bayesian neural networks learned with VB (in contrast to MAP estimation) due to the entropy term of the posterior approximation. We found the method described in the supplementary material to perform more robust than using e.g. the standard normal across different datasets.
>
> - This is a good suggestion. We added experiments on 2 datasets (UCI Energy, UCI Concrete), where we compare GRS to versions of GRS where i) the memory update is replaced by the k-center method and ii) where the Gaussian update is replaced by an optimisation of the ELBO with Dtk U Mtk \ Mtk-1 (as in VCL). Both approaches perform worse than GRS with both proposed components. The latter performs especially bad. This is because refitting requires optimising the ELBO with a small dataset. As we have pointed out in Sec. 3.3 (cf. Fig. 3), Bayesian neural networks with VB perform bad on small datasets due to over-regularisation. We emphasized this in the paper.
>
> - We agree that the paper is quite long and that it would be nice to include the results from the Appendix in the main text. However, all sections are already quite compressed and making more space necessarily comes with a compromise. The only possible option to shorten the OU process section would be to remove Fig. 2 from the main text. However, we think that these plots are very useful to understand and get an intuition for both approaches (BF vs. OU).
>
>
> Regarding the minors:
> - This is a good point, we now formulate it more precisely as "If the likelihood p(d|w) is close to the Gaussian r(w;d) in expectation w.r.t. the approximate posterior q(w)".
>
> - The Monte Carlo estimation refers to the estimation of the natural parameters in Eqs. (5a), (5b) (see also App. C). The two options for updating the Gaussian involve a sum of the natural parameter estimates, however with a different number of summands. Hence, we choose the option with the least summands. We clarified this in the revision of the paper.
>
> - While the experiments evaluating the memory in the online setting (Sec. 6.1) sample the data i.i.d. (random shuffling), the datasets used for evaluating the adaptation method (Sec. 6.2, see e.g. Figs. 5 and 6, and Fig. 9 in the supplementary material) are ordered and exhibit concept drift. The scenario the reviewer mentions can be seen as specific type of concept drift.

---

### Official Review · AnonReviewer1 · 2019-10-27
**Official Blind Review #1**

**Rating:** 6

**Review:**

This paper proposes a method for training Bayesian neural nets on a stream of non-stationary data. The authors propose a new way to approximate the variational distribution in the case where a memory (of past data) is used. The authors also use two forgetting mechanisms that allow the model to adapt to the changes in the data distribution.

Note that contrary to many recent works on continual learning that focus on not forgetting previous distributions, the goal here is to predict the current data.

I found this paper to be well written, the contribution is clear and the background material is well explained. The task is also important. I am not sure about the significance of the proposed approach. The motivation for providing a new memory update that takes into account the approximation error (from the variational approximation) is clear but I was not sure of the efficacy of the approach. Experimental results (Table 1) also show modest gains.  Experimental results using the adaptive method (Figure 5 & 6) are somewhat similar (e.g., why are other benchmarks not studied? gains are also modest).

- In Table 1, what does "bolding" indicate?
- In terms of other benchmarks, I was curious as to why you did not compare to "The Population Posterior and Bayesian Modeling on Streams" NIPS'15. As far as I understand this is also a Bayesian method that can accommodate non-stationary streams.
- VLC is proposed for the case of continual learning without forgetting.  It seems like you could study your method in such a setting (e.g., using online multi-task learning).
- In Section 4 (p.5 2nd paragraph), you mention that you assume that task boundaries are given. This seems to be a strong assumption and a significant change wrt to previous work. Could you perhaps add a practical example of this setup?
- I think it would be worthwhile to provide the full proposed algorithm somewhere in the paper and also discuss its computational complexity explicitly.

**Experience Assessment:**

I have read many papers in this area.

**Review Assessment: Checking Correctness Of Derivations And Theory:**

I assessed the sensibility of the derivations and theory.

**Review Assessment: Checking Correctness Of Experiments:**

I assessed the sensibility of the experiments.

**Review Assessment: Thoroughness In Paper Reading:**

I read the paper at least twice and used my best judgement in assessing the paper.

---

> ### Author Response · Authors · 2019-11-14
> **Response to reviewer questions**
>
> We thank the reviewer for the kind review and suggestions. We address each question in the given order.
>
> - In Table 1, bolding indicates the best results obtained by averaging over 16 independent runs with different random seeds and dataset shuffles. We used the same random shuffles for all competing methods. The random dataset shuffling introduces a high variance for all methods, and that is the reason why the differences seem statistically insignificant. A second source of randomness comes from the SGD/MC. We evaluated the number of runs (out of 16 for each) in which GRS performed better than all competing methods with the same random dataset shuffling. These numbers are reported below for each dataset and every memory size:
> UCI Energy: [10, 10, 15, 15, 13]
> UCI Concrete: [16, 15, 16, 16, 16]
> UCI Boston: [9, 11, 7, 5, 5]
> UCI Yacht: [12, 15, 15, 15, 16, 16]
> UCI Spam: [10, 8, 10, 10, 11]
> UCI Wine: [10, 11, 11, 13, 13]
> MNIST: [7, 12, 13, 12, 12]
>
> - Thank you for pointing out the paper "The Population Posterior and Bayesian Modeling on Streams", we were not aware of this work. The paper presents learning from streaming data by using a hierarchical model with a fixed prior (only the data distribution changes in time) by online learning on the parameters of the global variables. All past data is summarised by the posterior approximation (global variables) and forgotten/adapted through gradient based online learning. In our context (no local variables, only global) this would mean having a fixed prior, no memory, and doing only one gradient for \tilde{q}_{\theta__{t_k}} in Eq. (10) at each time-step t_k. We believe that having an evolving prior and a well defined forgetting mechanism gives us a better control over the learning process than the approach presented in the cited paper. We added this paper to the related work section.
>
> - This is an interesting suggestion. We did not consider our approach for online multi-task learning since a large part of our contribution is not suitable for this scenario: Our adaptation methods (Sec. 4) deliberately forget previous information (tasks) in order to adapt to the current data distribution (task). On the other hand, online multi-task learning is not formulated as learning from non-stationary data---all tasks are (equally) important and the model is often extended by task-dependent parameters.
> But we agree that our memory and Gaussian update (Sec. 3) could be used in this scenario with some modifications. In the multi-task experiments of VCL, the authors select a coreset from each task without ever evicting data from the coreset (moving it to the variational distribution). Said differently, the coreset grows (linearly) with the number of tasks. In contrast, our memory is taylored towards improving the posterior approximation with a fixed budget of data points in the running memory. To make a fair comparison, we would have to turn our running memory into coresets by "freezing" the coresets corresponding to old tasks, therefore preventing old data from being evicted. We consider this outside the scope of this work, although this an interesting future research direction.
>
> - Consider the special case where all datasets Dt are just a single data sample. In this case there are no task boundaries. For practical reasons (computational efficiency on GPUs), we want to infer the posterior from multiple data samples in parallel. We therefore assume that there is no drift within a short time-span.
>
> - This is a good suggestion, we added the algorithm for GRS + Bayesian forgetting and a short discussion regarding the computational complexity to the supplementary (cf. App. G).

---

> > ### Comment · AnonReviewer1 · 2019-11-15
> > **Thank you!**
> >
> > Thanks for your detailed reply and for providing an updated version of the paper (I think the algorithm in pseudo-code will be quite useful for readers). You answered all of my questions.

---

### Author Response · Authors · 2019-11-14
**Authors summary of updates to the paper**

We thank the reviewers for their polite and insightful comments. We updated the paper as suggested, the major changes are as follows:
- Added a pseudo algorithm and a discussion about its computational complexity (cf. App. G).
- Added experiments (on 2 datasets), evaluating the two components of GRS separately (cf. Fig. 8, App. A).
- Added a visualisation of the running memory at different time-steps on MNIST (cf. Fig. 9, App. A).

---

### Decision · Program_Chairs · 2019-12-19

**Decision:**

Accept (Poster)

**Comment:**

This paper introduces an algorithm for online Bayesian learning of both streaming and non-stationary data.  The algorithmic choices are heuristic but motivated by sensible principles.  The reviewers' main concerns were with novelty, but because the paper was well-written and addressing an important problem they all agreed it should be accepted.